# MTSTRec: Multimodal Time-Aligned Shared Token Recommender

**Ming-Yi Hong** [* 1]  **Yen-Jung Hsu** [* 2]  **Miao-Chen Chiang** [1]  **Che Lin** [1 2 3]

## Abstract

Sequential recommendation in e-commerce utilizes users' anonymous browsing histories to personalize product suggestions without relying on private information. Existing item ID-based methods and multimodal models often overlook the temporal alignment of modalities like textual descriptions, visual content, and prices in user browsing sequences. To address this limitation, this paper proposes the Multimodal Time-aligned Shared Token Recommender (MTSTRec), a transformer-based framework with a single time-aligned shared token per product for efficient cross-modality fusion. MTSTRec preserves the distinct contributions of each modality while aligning them temporally to better capture user preferences. Extensive experiments demonstrate that MTSTRec achieves state-of-the-art performance across multiple sequential recommendation benchmarks, significantly improving upon existing multimodal fusion. Our code is available at https://github.com/idssplab/MTSTRec.

## 1. Introduction

In e-commerce and online platforms, Sequential Recommendation Systems (SRS) play a pivotal role in delivering personalized product suggestions by analyzing users' browsing histories. Throughout its gradual evolution, SRS has transitioned from traditional probabilistic models to advanced neural networks, enabling more effective modeling of user behavior sequences. Recent Transformer-based methods, such as SASRec (Kang & McAuley, 2018) and BERT4Rec (Sun et al., 2019), have further enhanced sequence modeling by capturing dependencies across the entire interaction history. However, despite these advancements, existing SRS approaches largely focus on single-modal data, overlooking the potential of rich multimodal information to reveal deeper user preferences and improve recommendation quality.

Multimodal recommendation systems combine diverse data types, such as images and texts. Image-based methods use pre-trained convolutional neural networks (O'Shea, 2015) like ResNet (He et al., 2016) to capture the visual features, while text-based methods use models like BERT (Devlin et al., 2019) for product descriptions and reviews. CLIP aligns text and images through a multimodal approach, enabling zero-shot transfer for various vision-language tasks (Radford et al., 2021). Recently, large language models (LLMs) have excelled in extracting hidden textual information, enriching item representations (Geng et al., 2022; Lyu et al., 2024). Rather than full-scale LLM training, we focus on leveraging LLMs to enrich item descriptions by extracting hidden textual features. While many recommendation models rely on image classification or recognition to predict purchase intent, we argue that textual data alone is often sufficient for identifying what a product is. Instead, images should capture product aesthetics and style, which are especially important on platforms that sell the same product in various patterns or designs (Ugurlu, 2023).

When it comes to multimodal fusion, designing a unified model that effectively integrates different modalities presents significant challenges. Different modalities and their specialized formats learn at different speeds and patterns. This makes it challenging to combine them effectively into a single system. In the context of recommendation systems, the common modalities are typically images and text, which have fundamentally different input representations. Early fusion techniques are often used to unify these features, where modalities are combined before entering the recommendation model (He & McAuley, 2016; Liu et al., 2019). However, this approach fails to account for the significant differences in input representations and neural network architectures across modalities. In contrast, late fusion processes each modality independently before combining outputs later, enabling modality-specific extraction but ignoring their complementarity in representing the same product at each time step (Liang et al., 2023). This delays cross-modal interaction, forcing post hoc inference and leading to suboptimal representations.

To address both the modality processing and fusion issues, we propose **Multimodal Time-aligned Shared Token Rec-**

---

[*]Equal contribution [1]Data Science Degree Program, National Taiwan University and Academia Sinica, Taiwan [2]Graduate Institute of Communication Engineering, National Taiwan University, Taiwan [3]Department of Electrical Engineering, National Taiwan University, Taiwan. Correspondence to: Che Lin <che-lin@ntu.edu.tw>.

*Proceedings of the 42nd International Conference on Machine Learning*, Vancouver, Canada. PMLR 267, 2025. Copyright 2025 by the author(s).

ommender (**MTSTRec**), a novel framework for multimodal feature integration and cross-modal interaction using time-aligned shared tokens. MTSTRec consists of two main components: the Feature Extractor and the Multimodal Transformer. The feature extractor takes the browsing history sequence, where each item includes a product ID, image, text, and price. We use different extractors for each modality. For text, we enrich the data through LLMs with task-specific prompts to extract implicit consumer preferences. For images, we focus on style rather than classification, using Gram metrics to capture visual patterns that influence purchasing behavior. These inputs are then projected into separate feature embeddings for each modality. In the proposed multimodal transformer, a self-attention encoder is first applied to model the information for each feature. During fusion, we adopt a mid-fusion strategy, where modalities are processed independently and then combined in the intermediate stage. Our proposed Time-aligned Shared Token fusion module (**TST**) learns cross-modal interactions by aligning features from different modalities at each time step of the product interaction sequence. This ensures efficient feature sharing across modalities while maintaining the chronological order and consistency of product interactions.

Our contributions can be summarized as follows:

- We propose a unified multimodal recommendation framework for multimodal recommendations that seamlessly integrates diverse information, such as ID, text, image, and other modalities, enhancing the system's adaptability across different tasks.

- We introduce a novel **TST** module that leverages shared tokens to learn cross-modal interactions at each time step of the sequence, ensuring time-consistent alignment and fusion of modality information. By maintaining the chronological structure of the sequential data, the TST module effectively captures the evolving relationships between user interactions and product features.

- **MTSTRec** outperforms state-of-the-art methods on three real-world e-commerce datasets, setting a new benchmark in multimodal recommendation systems. Our ablation studies highlight the contribution of individual features and reveal deeper insights into the distinct roles of various modalities across diverse e-commerce environments, offering a valuable understanding of future advancements in the field.

## 2. Related Work

### 2.1. Sequential Recommendation Systems

SRS aims to predict the next item a user will interact with based on their browsing history, providing personalized recommendations. The traditional Markov Chain model (Shani et al., 2005) employed simple probabilistic methods but struggled to capture complex user behavior patterns. GRU4Rec (Hidasi, 2015) introduced RNNs to improve sequence modeling by capturing temporal dependencies. Transformer-based models like SASRec (Kang & McAuley, 2018) enhance this approach by using self-attention mechanisms (Vaswani, 2017) and causal masking, preserving temporal order and efficiently capturing short and long-term dependencies. BERT4Rec (Sun et al., 2019) extends this further by adopting a bidirectional Transformer model and using a CLOZE task for training. Unlike SASRec's causal masking, BERT4Rec allows the model to attend to both past and future items in the sequence, capturing richer contextual information. Transformers4Rec (Moreira et al., 2021) and GFormer (Li et al., 2023a) have been proposed to extend Transformer-based recommendations. Transformers4Rec adapts NLP Transformers to sequential recommendation tasks, whereas GFormer enhances user-item modeling by integrating generative self-supervised learning with a graph transformer architecture. However, these models remain focused primarily on single-modal data, overlooking the potential benefits of multimodal information.

### 2.2. Multimodal Recommendation Systems

Multimodal recommendation systems are essential for integrating information from multiple modalities to create more accurate and comprehensive predictions. These systems generally operate through two main stages: raw feature extraction and feature fusion (Liu et al., 2024).

Different modalities require tailored methods to capture their unique attributes in the raw feature extraction stage. For instance, image-based features are often extracted using CNNs (O'Shea, 2015) or, more recently, Vision Transformers (ViT) (Dosovitskiy et al., 2021), which excel at processing visual data. Textual features often rely on pre-trained language models (Devlin et al., 2019), capturing semantic meanings from product textual information. In addition, advances in LLMs (Achiam et al., 2023; Dubey et al., 2024) have significantly enhanced text-based feature extraction. LLM-Rec (Lyu et al., 2024) utilizes LLM and prompts to generate richer contextual representations, enhancing recommendation quality. Other LLM-based approaches have also emerged, further refining text comprehension to optimize the use of textual features in recommendation systems (Zhao et al., 2023; Li et al., 2023b; Wu et al., 2024).

After raw feature extraction, the system proceeds to the feature fusion stage, where the multimodal data are combined and processed. It can be categorized into three main approaches (Zhou et al., 2023): (i) Early fusion involves merging the features from different modalities at the initial stages of the model. For example, VBPR (He & McAuley,

2016) integrates visual features into a matrix factorization approach. However, early fusion may miss important temporal relationships and modality-specific behaviors by merging features too early. (ii) Mid-fusion delays modality combination for more refined processing, as seen in MM-Rec (Wu et al., 2021), which uses a cross-modal attention mechanism to combine textual and visual information effectively. However, these methods often fail to capture temporal relationships, which are crucial in sequential recommendations. (iii) Late fusion keeps modalities separate until the final stage, as in MMMLP (Liang et al., 2023), the final outputs from three different modalities are concatenated before making the prediction. While preserving modality-specific features, it fails to capture early interactions between modalities and sequences, leading to suboptimal performance with complex sequential patterns. Similarly, FREEDOM (Zhou & Shen, 2023) freezes item-item structures and denoises user-item interactions before fusion, while Mirror Gradient (Zhong et al., 2024) improves robustness via flat local minima exploration, both maintaining separate modality processing until the final stage.

Recent works also emphasize robustness and information efficiency during fusion. CGI (Wei et al., 2022) enhances recommendation systems through contrastive graph learning, combining adaptive structure pruning with the information bottleneck to suppress irrelevant signals. DVIB (Zhao et al., 2025) improves multimodal recommendation by applying hidden-layer perturbations and self-distillation, inducing an information bottleneck effect. These approaches conceptually align with our mid-fusion strategy, which aims to preserve relevant information during fusion.

### 2.3. Temporal Dynamics in Recommendation Systems

Recent research highlights the significance of temporal dynamics in recommender systems. For instance, time-aware recommendation systems (TARS) have been emphasized (Campos et al., 2014). Temporal and feature dynamics were integrated to address data sparsity (Zhang et al., 2021), while multi-scale temporal effects were leveraged for micro-video recommendations (Jiang et al., 2020). CDTR (Wang et al., 2024) introduced a causality-based framework that addresses item- and time-level biases in user behavior to enhance the accuracy of time-aware recommender models.

While these approaches explicitly model timestamps or temporal dependencies, our work focuses on a different aspect of time representation. Instead of relying on absolute timestamps, we consider the relative positional structure of different modalities within a sequence. This allows for a structured alignment of multimodal data without requiring explicit time information. Our method does not predict future interactions based on timestamp trends but ensures that multimodal features at each step are temporally aligned,

preserving the chronological order of product interactions.

## 3. Proposed Method: MTSTRec

This section presents the MTSTRec framework, designed to enhance multimodal sequential recommendation by effectively fusing diverse product modalities from a consumer-centric perspective. The framework comprises two primary modules: the feature extractor module, which processes browsing history to extract modality-specific embeddings, and the multimodal transformer with time-aligned shared token (TST) fusion, which synchronizes and integrates features across modalities and layers. An overview of the MTSTRec is illustrated in Figure 1.

### 3.1. Preliminaries

Let the set of items be defined as $\boldsymbol{I} = \{\boldsymbol{i}_1, \boldsymbol{i}_2, \dots, \boldsymbol{i}_k, \dots, \boldsymbol{i}_{|I|}\}$, where each item $\boldsymbol{i}_k \in \boldsymbol{I}$ consists of four essential elements: product ID $\boldsymbol{b}_k$, image $\boldsymbol{v}_k$, text $\boldsymbol{t}_k$, and price $\boldsymbol{c}_k$. Therefore, each item $\boldsymbol{i}_k$ can be expressed as a tuple $\boldsymbol{i}_k = (\boldsymbol{b}_k, \boldsymbol{v}_k, \boldsymbol{t}_k, \boldsymbol{c}_k)$, capturing the multiple modalities that describe it. For each user, their browsing history is denoted as $\boldsymbol{S} = [\boldsymbol{s}_1, \boldsymbol{s}_2, \dots, \boldsymbol{s}_n]$, where each $\boldsymbol{s}_i \in \boldsymbol{I}$ represents an item the user has engaged with, and $n$ indicates the session length. The goal of SRS is to predict the next item $\boldsymbol{s}_{n+1}$ that a user will engage with based on their interaction history $\boldsymbol{S}$.

### 3.2. Feature Extractors

#### 3.2.1. ID EXTRACTOR

The ID extractor converts raw ID data into meaningful embeddings. We construct an item ID embedding matrix $\boldsymbol{M}^{\text{id}} \in \mathbb{R}^{d \times |I|}$, where $d$ denotes the dimension of the ID embedding. The input ID embedding matrix for each session $\boldsymbol{E}^{\text{id}} \in \mathbb{R}^{d \times n}$ is then retrieved such that $\boldsymbol{E}^{\text{id}}_i = \boldsymbol{M}^{\text{id}}_{\boldsymbol{s}_i}$, corresponding to the item $\boldsymbol{s}_i$ in the user's interaction sequence, the $i$-th row in the matrix. For any padding items in the sequence, a constant zero vector $\boldsymbol{0}$ is used as their embedding, typically added when sequences are shorter than the maximum length. This mechanism allows the model to capture item relationships and improve recommendation accuracy by understanding item identities within sequences. The processing method for the price extractor is similar; refer to Appendix A for more details.

#### 3.2.2. STYLE EXTRACTOR

We are inspired by the Neural Style Transfer algorithm (Gatys, 2015; Gatys et al., 2016) to extract style features. Instead of transferring style, we focus solely on extracting it. Using the VGG-19 (Simonyan & Zisserman, 2015) model, we pass the image through the network to obtain feature maps, from which we compute Gram matrices $\boldsymbol{G}$. These

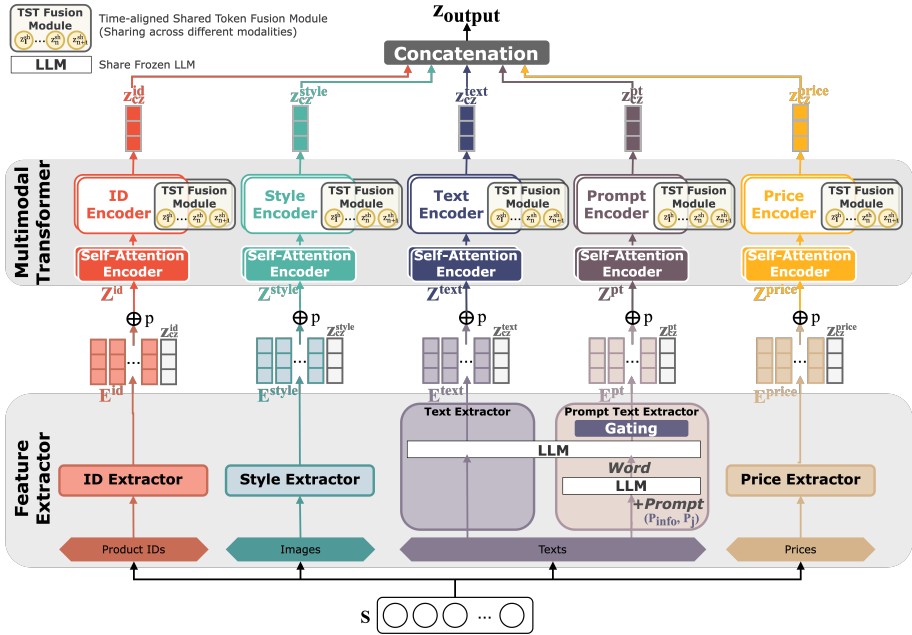

*Figure 1.* MTSTRec consists of feature extractors and multimodal transformers. The feature extractors take inputs as a browsing history sequence $\boldsymbol{S}$, where each item consists of a product ID, image, text, and price, transforming them into the corresponding feature embeddings, along with a CLOZE embedding $\boldsymbol{z}_{\mathrm{cz}}$ for each modality. The multimodal transformers begin by applying a self-attention encoder to independently process the information for each feature. When fusion begins, we utilize the TST module across different modalities at each time step and across multiple layers, ensuring both modality-specific and cross-modal information are captured effectively.

matrices capture image texture and color patterns and are invariant to spatial transformations, ensuring similar matrices for images with similar styles. We then generate the input style embedding matrix for each session, $\boldsymbol{E}^{\mathrm{style}} \in \mathbb{R}^{d \times n}$. Further details are in the Appendix B.

### 3.2.3. TEXT EXTRACTOR

The text extractor focuses on extracting features from product titles and descriptions. We use Llama 3.1 (Dubey et al., 2024) as our backbone for generating text embeddings that capture the key attributes of the product (see Appendix C for details). Each product is assigned a text embedding, forming the item text embedding matrix $\boldsymbol{M}^{\mathrm{text}} \in \mathbb{R}^{d \times |I|}$, which is generated by feeding the textual information of the product into an LLM text encoder to obtain the initial text embeddings, followed by a one-layer projection to adjust the dimensionality. The input text embedding matrix for each session $\boldsymbol{E}^{\mathrm{text}} \in \mathbb{R}^{d \times n}$ is then retrieved, such that $\boldsymbol{E}_i^{\mathrm{text}} = \boldsymbol{M}_{\boldsymbol{s}_i}^{\mathrm{text}}$ corresponding to the item $\boldsymbol{s}_i$ in the user's interaction sequence.

### 3.2.4. PROMPT-TEXT EXTRACTOR

In the second part of text processing, we explore how LLMs can be prompted to generate additional textual information

to improve recommendation performance. Inspired by LLM-Rec (Lyu et al., 2024), we employ two prompt strategies: basic prompt and recommendation prompt. Additionally, we utilize five variations tailored to the task, including three for basic tasks, along with two designed for recommendation tasks (see Appendix D for more details). The LLM input is divided into a system prompt and a user prompt. The system prompt provides a brief description of the e-commerce context $P_{\mathrm{info}}$ to ensure that the LLM understands the task's background, and a task-specific prompt $P^j$, selected from the five predefined prompt variations described earlier. For each product $i$, its textual information, such as the title and description, is fed into the user prompt. The LLM then generates a response $\boldsymbol{P}_i^j$ for each product $i$, using each prompt variant $j$. These responses are converted into prompt embeddings for further processing (see Appendix D for more details). The process is formalized as follows:

$$\hat{\boldsymbol{P}}_i^j = Encoder_{LLM}(\boldsymbol{P}_i^j). \tag{1}$$

To further enhance the model's capacity to integrate and utilize the generated prompt embeddings, we introduce a gating mechanism. Inspired by (Yu et al., 2024), the gating network $G$ is designed to manage the varying importance of each prompt variant $\hat{\boldsymbol{P}}^j$. This network controls the information flow to the final prediction layer. The weights of the

prompt-text embeddings $\boldsymbol{w}$ are calculated as follows:

$$\boldsymbol{w} = G(\|_j \hat{\boldsymbol{P}}^j) := softmax(W[\|_j \hat{\boldsymbol{P}}^j] + b), \\ \forall j \in \{1, 2, \ldots, 5\}. \quad (2)$$

The item prompt embedding matrix $\boldsymbol{M}^{\text{pt}} \in \mathbb{R}^{d \times |I|}$ is then computed by:

$$\boldsymbol{M}^{\text{pt}} = L_p \left( \sum_{j=1}^{5} \boldsymbol{w}_j \cdot \hat{\boldsymbol{P}}^j \right), \quad (3)$$

where $L_p$ is a linear projection layer applied to the gated prompt embeddings. Further details are provided in the Appendix E. The input prompt embedding matrix for each session $\boldsymbol{E}^{\text{pt}} \in \mathbb{R}^{d \times n}$ is then retrieved, such that $\boldsymbol{E}_i^{\text{pt}} = \boldsymbol{M}_{\boldsymbol{s}_i}^{\text{pt}}$ matching item $\boldsymbol{s}_i$ within the user's interaction sequence.

### 3.3. Multimodal Transformer with Time-aligned Shared Token Fusion

The multimodal transformer is built based on the Transformer encoder architecture (Vaswani, 2017), where each feature has its own encoder, to begin with. The user sequence is processed through each feature extractor, producing five distinct embeddings: $\boldsymbol{E}^{\text{id}}, \boldsymbol{E}^{\text{style}}, \boldsymbol{E}^{\text{text}}, \boldsymbol{E}^{\text{pt}}, \boldsymbol{E}^{\text{price}}$, all in $\mathbb{R}^{d \times n}$. Positional embeddings are added to each embedding, and a CLOZE embedding $\boldsymbol{z}_{\text{cz}}$ is added to the sequence to signify the item to be predicted. The recommendation is made based on the final representation of $\boldsymbol{z}_{\text{cz}}$. Using the ID embedding matrix $\boldsymbol{E}^{\text{id}}$ as an example, the final input to the ID encoder is:

$$\boldsymbol{Z}^{\text{id}} = [\boldsymbol{E}_1^{\text{id}}, \boldsymbol{E}_2^{\text{id}}, ..., \boldsymbol{E}_N^{\text{id}}, \boldsymbol{z}_{\text{cz}}^{\text{id}}] \oplus \boldsymbol{p}, \quad (4)$$

where $\boldsymbol{p} \in \mathbb{R}^{d \times (n+1)}$ is the learnable positional embedding, and $\oplus$ denotes element-wise addition. Thus, we obtain the inputs to the multimodal transformer: $\boldsymbol{Z}^{\text{id}}, \boldsymbol{Z}^{\text{style}}, \boldsymbol{Z}^{\text{text}}, \boldsymbol{Z}^{\text{pt}}, \boldsymbol{Z}^{\text{price}}$, which are passed through separate encoders.

#### 3.3.1. SELF-ATTENTION ENCODER

In the next step, input embeddings from each modality $\boldsymbol{Z}$ are processed independently through a self-attention encoder (Vaswani, 2017). This approach enables each modality to learn its unique features, and the resulting representations are then prepared for cross-modal fusion using the TST module. Further details are provided in Appendix F.

#### 3.3.2. TIME-ALIGNED SHARED TOKEN FUSION WITH SEQUENTIAL MULTIMODAL INTEGRATION

Our TST fusion module, as illustrated in Figure 2, facilitates multimodal information sharing by aligning item tokens across different modalities. Inspired by the attention bottleneck mechanism (Nagrani et al., 2021), which focuses on

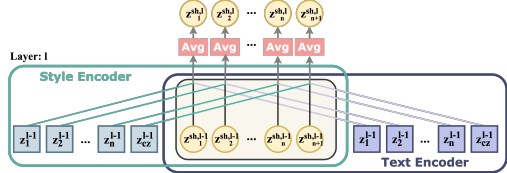

*Figure 2.* The TST module is shared across different modalities in a manner that aligns each item in the sequence. For example, in the fusion of style and text embeddings, each TST embedding $\boldsymbol{z}^{\text{sh}}$ is updated element-wise by averaging the corresponding tokens from the style and text modalities at the same time step in the sequence, ensuring cross-modal information exchange that aligns in time.

efficiently transferring information between modalities, we re-design the mid-fusion module such that the embeddings for each item are aligned in time. This design introduces a strong prior that the embeddings from each modality belong to the same item in the sequence, ensuring that representations of the same item from different modalities are effectively fused.

Each sequence contains $n + 1$ tokens, including the CLOZE embedding $\boldsymbol{z}_{\text{cz}}$ (Sun et al., 2019). The number of time-aligned shared tokens ($\boldsymbol{z}^{\text{sh}}$) matches the sequence length, with each shared token $\boldsymbol{z}_t^{\text{sh}}$ at time step $t$ corresponding to modality-specific tokens $\boldsymbol{z}_t^{\text{mod}}$, where $\text{mod} \in \{\text{id, style, text, pt, price}\}$. These shared tokens enable cross-modal interaction for the same item at each time step. For each modality, modality-specific tokens $\boldsymbol{Z}^{\text{mod}} = [\boldsymbol{z}_1^{\text{mod}}, \ldots, \boldsymbol{z}_n^{\text{mod}}, \boldsymbol{z}_{\text{cz}}^{\text{mod}}]$ and the time-aligned shared tokens $\boldsymbol{Z}^{\text{sh}} = [\boldsymbol{z}_1^{\text{sh}}, \ldots, \boldsymbol{z}_n^{\text{sh}}, \boldsymbol{z}_{n+1}^{\text{sh}}]$ are concatenated and fed into the self-attention encoder $Att$ (Vaswani, 2017). It processes input tokens from each modality independently through $L$ transformer layers, where the transformation at layer $l$ is defined as:

$$\boldsymbol{Z}^{\text{mod},l} = Att(\boldsymbol{Z}^{\text{mod},l-1} \| \boldsymbol{Z}^{\text{sh},l-1}). \quad (5)$$

At each layer, modality-specific tokens are updated into $\boldsymbol{Z}^{\text{mod},l}$ for the next layer $l$. These tokens capture the modality-specific features, which will be further enhanced by interaction with the TST module. Similarly, for time-aligned shared tokens, the tokens from all modalities are averaged at the end of each layer to form the next layer's shared tokens $\boldsymbol{Z}^{\text{sh},l}$, facilitating further fusion across subsequent layers. This process is defined as:

$$\boldsymbol{Z}^{\text{sh},l} = \frac{1}{\#\text{mod}} \sum_{\text{mod}} Att(\boldsymbol{Z}^{\text{mod},l-1} \| \boldsymbol{Z}^{\text{sh},l-1}), \\ \forall \text{mod} \in \{\text{id, style, text, pt, price}\}. \quad (6)$$

In the first layer of the fusion encoder, the modality-specific sequences do not take inputs from the time-aligned shared

tokens yet, as these tokens are initially unlearned. However, each shared token $z^{\text{sh}}$ can take inputs from its corresponding modality tokens, learning multimodal information. From the second layer onward, modality-specific tokens and their corresponding shared tokens attend to each other, enabling cross-modal learning and information exchange.

After fusion, the learned representations are passed to the prediction layer. The sequence token $z_{\text{cz}}$ plays a critical role in capturing the information needed for predicting the next item in the user's interaction sequence. By integrating information from the modality-specific and time-aligned shared tokens, the model achieves a cohesive multimodal representation, improving the accuracy of the recommendation task. The final modality representations are normalized to ensure stability, then combined and used for prediction and loss computation. The $z_{\text{cz}}$ tokens are concatenated to form the final recommendation representation for each sequence:

$$z_{\text{output}} = z_{\text{cz}}^{\text{id}} \| z_{\text{cz}}^{\text{style}} \| z_{\text{cz}}^{\text{text}} \| z_{\text{cz}}^{\text{pt}} \| z_{\text{cz}}^{\text{price}}. \tag{7}$$

This final representation $z_{\text{output}}$ encapsulates the key information from each modality, allowing the model to make accurate predictions for the next item in the user interaction sequence.

### 3.4. Loss Function

For each sequence, the model computes the cosine similarity between the final recommendation representation $z_{\text{output}}$ and both the ground truth and negative samples. The embeddings for both the ground truth and negative samples $y$ are generated by concatenating features $E$ extracted for each item through the same feature extractors. Let $\cos(z_{\text{output}}, y^k)$ represent the cosine similarity between $z_{\text{output}}$ and the $k$-th sample, where $k \in K_{gt}$ for positive samples (ground truth) and $k \in K_n$ for negative samples. These cosine similarities are then employed to compute the binary cross-entropy (BCE) loss for the sequence as follows:

$$\mathcal{L}_{\text{BCE}} = -\frac{1}{|K_{gt} \cup K_n|} \Bigg( \sum_{k \in K_{gt}} \log\big(\cos(z_{\text{output}}, y^k)\big) + \\ \sum_{k \in K_n} \log\big(1 - \cos(z_{\text{output}}, y^k)\big) \Bigg). \tag{8}$$

This formula drives the model to maximize cosine similarity with ground truth and minimize it with negative samples, improving recommendation performance.

## 4. Experiments

### 4.1. Experimental Settings

**Dataset.** Our experiments utilize three datasets: two proprietary datasets from AviviD Innovative Multimedia—*Food*

*E-commerce* and *House-Hold E-commerce*, which have already been made publicly available[1], and one public dataset from *H&M*. The proprietary datasets capture user interactions, including page views and purchases, while the public dataset focuses on purchase records in the trousers category from *H&M*. Detailed information, including dataset splitting, is provided in Appendix G.

**Evaluation Metrics.** We evaluate all models using three popular top-k ranking metrics: Normalized Discounted Cumulative Gain (NDCG@k), Hit Rate (HR@k), and Mean Reciprocal Rank (MRR@k), with k set to 5 and 10. NDCG@k remains unchanged as it inherently supports multiple selections. For HR@k and MRR@k, we adjust the calculations to account for the multiple-answer format, ensuring a fair evaluation. For more detail, please refer to Appendix H.

**Benchmark Models.** We compare our MTSTRec model with two categories of baselines: general models using only item IDs (e.g., SASRec (Kang & McAuley, 2018) and BERT4Rec (Sun et al., 2019)) and multimodal models (e.g., MMMLP (Liang et al., 2023)). The latter also includes the enhanced versions of SASRec$^+$ and BERT4Rec$^+$, which concatenate image and text features, similar to our model. These modifications allow them to act as early fusion models, combining all features upfront to ensure a fair comparison of multimodal inputs. For more details on the benchmark models, refer to Appendix I.

### 4.2. Performance Comparison

To assess the generalizability of MTSTRec, we conducted experiments on three datasets and compared the results with baseline models. More implementation details can be found in Appendix J. The results, summarized in Table 1, reveal critical insights into the effectiveness of different models.

MTSTRec surpasses all baselines, leveraging its TST module to outperform both early and late fusion approaches. In the *Food E-commerce* dataset, MTSTRec achieves the highest NDCG@5 score of 0.8800, and in the *House-Hold E-commerce* dataset, it achieves 0.8942, outperforming SASRec$^+$ by approximately 3.4% and 9.7%, respectively. Notably, in the *H&M (Trousers)* dataset, MTSTRec achieves an NDCG@5 of 0.2307, further solidifying its dominance across all metrics.

Besides, SASRec$^+$ and BERT4Rec$^+$, which integrate text and image features, further boost performance. In the *Food E-commerce* dataset, SASRec$^+$ achieves an NDCG@5 of 0.8512, and in the *House-Hold E-commerce* dataset, it reaches 0.8150, which demonstrates the importance of incor-

---

[1]The datasets are available at https://github.com/idssplab/MTSTRec. For more details regarding the dataset release, please refer to Appendix O.

*Table 1.* Performance comparison of benchmark models and MTSTRec on three datasets.

| DATASET | MODEL | NDCG@5 | NDCG@10 | HR@5 | HR@10 | MRR@5 | MRR@10 |
|---|---|---|---|---|---|---|---|
| *Food E-commerce* | SASREC | 0.8015 | 0.7999 | 0.7505 | 0.7836 | 0.7099 | 0.7143 |
| | BERT4REC | 0.8076 | 0.8094 | 0.7658 | 0.8010 | 0.7146 | 0.7193 |
| | SASREC$^+$ | 0.8512 | 0.8446 | 0.8013 | 0.8250 | 0.7729 | 0.7760 |
| | BERT4REC$^+$ | 0.8441 | 0.8394 | 0.7962 | 0.8218 | 0.7641 | 0.7675 |
| | MMMLP | 0.8276 | 0.8239 | 0.7845 | 0.8129 | 0.7450 | 0.7487 |
| | MMMLP$^+$ | 0.8164 | 0.8130 | 0.7742 | 0.8048 | 0.7338 | 0.7379 |
| | MTSTREC | **0.8800** | **0.8765** | **0.8407** | **0.8651** | **0.8086** | **0.8118** |
| *House-Hold E-commerce* | SASREC | 0.7563 | 0.7736 | 0.7578 | 0.7960 | 0.7024 | 0.7076 |
| | BERT4REC | 0.7596 | 0.7761 | 0.7603 | 0.7968 | 0.7068 | 0.7116 |
| | SASREC$^+$ | 0.8150 | 0.8258 | 0.8144 | 0.8410 | 0.7688 | 0.7723 |
| | BERT4REC$^+$ | 0.7969 | 0.8110 | 0.7959 | 0.8287 | 0.7477 | 0.7521 |
| | MMMLP | 0.8335 | 0.8425 | 0.8303 | 0.8525 | 0.7930 | 0.7959 |
| | MMMLP$^+$ | 0.8401 | 0.8480 | 0.8368 | 0.8569 | 0.7993 | 0.8019 |
| | MTSTREC | **0.8942** | **0.9086** | **0.9067** | **0.9358** | **0.8568** | **0.8607** |
| *H&M (Trousers)* | SASREC | 0.1520 | 0.1759 | 0.1809 | 0.2489 | 0.1258 | 0.1348 |
| | BERT4REC | 0.1468 | 0.1692 | 0.1738 | 0.2378 | 0.1223 | 0.1306 |
| | SASREC$^+$ | 0.1605 | 0.1811 | 0.1828 | 0.2415 | 0.1371 | 0.1448 |
| | BERT4REC$^+$ | 0.1633 | 0.1848 | 0.1878 | 0.2492 | 0.1387 | 0.1466 |
| | MMMLP | 0.1754 | 0.1923 | 0.1971 | 0.2451 | 0.1502 | 0.1565 |
| | MMMLP$^+$ | 0.1710 | 0.1889 | 0.1927 | 0.2436 | 0.1461 | 0.1528 |
| | MTSTREC | **0.2307** | **0.2797** | **0.3139** | **0.4481** | **0.1871** | **0.2049** |

porating multimodal data to better capture user preferences and provide more accurate recommendations. However, these models still fell short of MTSTRec's performance.

MMMLP$^+$ uses the same features as MTSTRec, incorporating additional features compared to MMMLP. While MMMLP$^+$ shows slight improvements over MMMLP in the *House-Hold E-commerce* dataset, it performs slightly worse in the other two datasets. This suggests that MMMLP is better suited for simpler inputs. When comparing MMMLP$^+$ with MTSTRec, MTSTRec demonstrates a superior ability to process and leverage the complex relationships between multiple features. As a result, MTSTRec achieves significantly better results across all three datasets, further highlighting its effectiveness in capturing multimodal interactions.

In summary, MTSTRec consistently outperforms both early fusion models (SASRec$^+$ and BERT4Rec$^+$) and the late fusion model (MMMLP) across all evaluated datasets. The model's effective integration of multimodal information via the TST module ensures that feature interactions are fully captured, leading to state-of-the-art performance, particularly in complex datasets like *H&M (Trousers)*, with performance gains of approximately 31.5%–43.7% over competing models in terms of NDCG@5.

### 4.3. Ablation Study of Modalities

In this section, we conduct an ablation study on *Food E-commerce* and *House-Hold E-commerce* datasets to evaluate the contribution of different modalities, including item ID, text, prompt text, images, and price. As shown in Table 2, removing the item ID modality has the most significant impact on both datasets, with NDCG@5 dropping sharply from

0.8800 to 0.7582 on the *Food E-commerce* dataset. This highlights the critical role of product identity in differentiating items. The absence of item IDs also increased training time as the model struggled to manage without clear product identifiers. See Appendix K for a detailed analysis of the impact of removing ID modules across different datasets. Text information proves to be another important modality. In the *House-Hold E-commerce* dataset, removing both text and prompt text leads to a noticeable performance drop, with NDCG@5 decreasing from 0.8942 to 0.8191. This highlights the significance of detailed product descriptions in capturing user preferences. Even when standard text is removed, the prompt text still provides valuable contextual information, enabling the model to maintain reasonable accuracy, as indicated by a smaller drop to 0.8488 NDCG@5 compared to removing both text and prompt text.

In summary, while item ID is crucial for accurate recommendations, our approach surpasses the ID-only baselines (SASRec and BERT4Rec) by demonstrating the added value of integrating multiple modalities. Our multimodal design provides complementary strengths beyond the limitations of ID-based methods. Interestingly, despite its relevance in consumer decisions, price contributes little to recommendation accuracy, suggesting that user preferences are better captured through semantically rich and behaviorally informative features. Our approach demonstrates the power of leveraging multimodal inputs to enhance accuracy and robustness in predicting user preferences.

To better understand how MTSTRec allocates attention across modalities, we visualize self-attention produced just before the final output layer of the TST module on the *House-Hold E-commerce* dataset (Figure 3). The attention maps reveal distinct modality-specific patterns. Product

*Table 2.* The impact of removing different modality modules across e-commerce platforms.

| ABLATION STUDY | Food E-commerce | | | House-Hold E-commerce | | |
|---|---|---|---|---|---|---|
| | NDCG@5 | HR@5 | MRR@5 | NDCG@5 | HR@5 | MRR@5 |
| MTSTREC | **0.8800**(±0.0023) | **0.8407**(±0.0031) | **0.8086**(±0.0023) | **0.8942**(±0.0035) | **0.9067**(±0.0024) | **0.8568**(±0.0041) |
| W/O ID | 0.7582(±0.0049) | 0.7506(±0.0049) | 0.6459(±0.0062) | 0.7913(±0.0244) | 0.8337(±0.0191) | 0.7184(±0.0288) |
| W/O TEXT & PROMPT | 0.8574(±0.0013) | 0.8102(±0.0011) | 0.7813(±0.0015) | 0.8191(±0.0047) | 0.8142(±0.0041) | 0.7761(±0.0062) |
| W/O TEXT | 0.8729(±0.0016) | 0.8308(±0.0024) | 0.7998(±0.0022) | 0.8488(±0.0095) | 0.8553(±0.0139) | 0.8051(±0.0087) |
| W/O PROMPT TEXT | 0.8749(±0.0046) | 0.8331(±0.0056) | 0.8030(±0.0054) | 0.8770(±0.0035) | 0.8895(±0.0079) | 0.8366(±0.0031) |
| W/O STYLE | 0.8784(±0.0020) | 0.8391(±0.0030) | 0.8068(±0.0026) | 0.8932(±0.0093) | 0.9061(±0.0082) | 0.8524(±0.0111) |
| W/O PRICE | 0.8791(±0.0025) | 0.8404(±0.0031) | 0.8077(±0.0030) | 0.8941(±0.0048) | 0.9066(±0.0040) | 0.8553(±0.0063) |

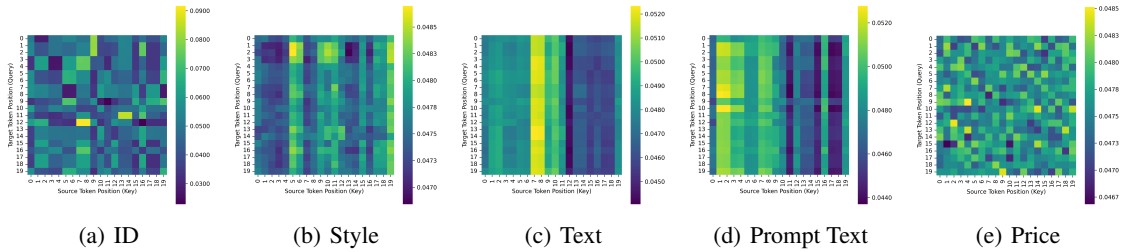

| (a) ID | (b) Style | (c) Text | (d) Prompt Text | (e) Price |

*Figure 3.* Self-attention heatmaps from the last TST layer of the MTSTRec model on *House-Hold E-commerce* dataset, visualized across five modalities. Each $20 \times 20$ heatmap shows attention weights from queries (y-axis) to keys (x-axis). Brighter regions indicate higher attention. The specific light and dark patterns highlight how the model allocates focus based on content.

ID and price tokens exhibit scattered, weakly structured attention, while style (image) features show broader focus with noticeable vertical stripes. Text and prompt text display strong vertical attention on some key tokens and share a consistent pattern—higher attention in the first half of the sequence (e.g., indices 1–10) and reduced focus in the latter half. These patterns indicate that text and prompt text convey semantically aligned and informative content. Complementary attention is also observed across modalities; for instance, at indices 5 and 6, style receives strong attention while text and prompt text are relatively weak. This highlights MTSTRec's ability to adapt its attention across modalities, reinforcing the importance of a multimodal design for recommendation.

### 4.4. Impact of Fusion Strategies

In this ablation study, we evaluated the impact of different fusion strategies on the *Food E-commerce* and *House-Hold E-commerce* datasets, comparing TST (used in MTSTRec) with two alternative setups: one resembling late fusion by removing the TST and fusion encoder; another resembling early fusion by removing the entire multimodal transformer. In the late fusion alternative, features are processed independently and concatenated only at the final step, and in the early fusion alternative, all features are concatenated immediately after feature extraction and passed through a single encoder.

TST consistently outperforms both alternatives as shown in Table 3. On the *Food E-commerce* dataset, TST achieves an NDCG@5 of 0.8800, compared to 0.8621 for early fusion and 0.8211 for late fusion. Similarly, on the *House-Hold E-commerce* dataset, TST obtains an even higher NDCG@5 of 0.8942, outperforming early fusion (0.8366) and late fusion (0.8773). These trends reflect two limitations of the baseline approaches. First, in early fusion, all features are processed together from the start, which can dilute unique information from each modality, as the model cannot treat them distinctly. Consequently, the model cannot fully leverage the strengths of each feature, leading to reduced performance. The performance gap is particularly noticeable in the *House-Hold E-commerce* dataset, where early fusion reduces the NDCG@5 by 0.0576 compared to TST. Second, late fusion limits the interaction between different features until the end of the sequence. This restriction hinders the model's ability to capture cross-modal dependencies throughout the sequence, leading to decreased performance. This limitation is especially evident in the *Food E-commerce* dataset, where the NDCG@5 score drops from 0.8800 to 0.8211, decreasing to 0.0589.

In conclusion, TST's ability to facilitate cross-modal interaction at each time step leads to better results, whereas early and late fusion approaches fall short due to inefficient handling of feature interactions.

### 4.5. Impact of Time-aligned Shared Tokens

We evaluate the different shared token configurations on the *Food E-commerce* dataset, focusing on three key setups: the proposed TST (1:1), TST with multiple shared tokens per

Table 3. The impact of removing different modules across e-commerce platforms.

| DATASET | FUSION METHOD | NDCG@5 | NDCG@10 | HR@5 | HR@10 | MRR@5 | MRR@10 |
|---------|---------------|--------|---------|------|-------|-------|--------|
| *Food E-commerce* | MTSTREC | **0.8800** | **0.8765** | **0.8407** | **0.8651** | **0.8086** | **0.8118** |
| | W/O TST & FUSION ENCODER (LATE FUSION) | 0.8211 | 0.8301 | 0.7912 | 0.8378 | 0.7271 | 0.7333 |
| | W/O MULTIMODAL ENCODER (EARLY FUSION) | 0.8621 | 0.8590 | 0.8206 | 0.8483 | 0.7862 | 0.7899 |
| *House-Hold E-commerce* | MTSTREC | **0.8942** | **0.9086** | **0.9067** | **0.9358** | **0.8568** | **0.8607** |
| | W/O TST & FUSION ENCODER (LATE FUSION) | 0.8773 | 0.8896 | 0.8839 | 0.9116 | 0.8392 | 0.8429 |
| | W/O MULTIMODAL ENCODER (EARLY FUSION) | 0.8366 | 0.8516 | 0.8404 | 0.8738 | 0.7929 | 0.7974 |

Table 4. The impact of time-aligned shared tokens on Food E-commerce dataset results.

| SETTING | NDCG@5 | NDCG@10 | HR@5 | HR@10 | MRR@5 | MRR@10 |
|---------|--------|---------|------|-------|-------|--------|
| TST (1:1) (OURS) | **0.8800**($\pm$0.0023) | **0.8765**($\pm$0.0021) | **0.8407**($\pm$0.0031) | **0.8651**($\pm$0.0029) | **0.8086**($\pm$0.0023) | **0.8118**($\pm$0.0023) |
| TST (1:2) | 0.8795($\pm$0.0022) | 0.8758($\pm$0.0020) | 0.8406($\pm$0.0025) | 0.8641($\pm$0.0023) | 0.8079($\pm$0.0026) | 0.8110($\pm$0.0026) |
| TST (1:4) | 0.8769($\pm$0.0041) | 0.8737($\pm$0.0038) | 0.8384($\pm$0.0039) | 0.8631($\pm$0.0036) | 0.8044($\pm$0.0051) | 0.8077($\pm$0.0050) |
| BOTTLENECKS (ALL:5) | 0.8737($\pm$0.0028) | 0.8695($\pm$0.0024) | 0.8323($\pm$0.0030) | 0.8560($\pm$0.0027) | 0.8012($\pm$0.0029) | 0.8044($\pm$0.0028) |
| BOTTLENECKS (ALL:21) | 0.8754($\pm$0.0036) | 0.8716($\pm$0.0035) | 0.8346($\pm$0.0043) | 0.8587($\pm$0.0040) | 0.8037($\pm$0.0039) | 0.8069($\pm$0.0038) |

time step (TST (1:2) and TST (1:4)), and fusion bottlenecks. Each configuration represents a unique mid-fusion approach to feature sharing during the sequence learning process. Further details are in the Appendix L.

As shown in Table 4, the proposed TST (1:1) configuration delivers the best performance, with an NDCG@5 of 0.8800. This setup ensures that only one shared token per time step facilitates information transfer, which helps the model focus on product-specific features without introducing unnecessary noise. TST (1:2) and TST (1:4) show a slight drop in performance, with NDCG@5 ranging from 0.8769 to 0.8795. The increase in shared tokens per time step introduces more information exchange but also adds redundancy and potential noise, slightly affecting model efficiency. Bottlenecks configurations perform worse than TST, with BOTTLENECKS (ALL:5) (Nagrani et al., 2021) achieving an NDCG@5 of 0.8737 and BOTTLENECKS (ALL:21) achieving an NDCG@5 of 0.8754, which is 0.5% lower than TST (1:1). Despite allowing broader information sharing, these setups lack the precise, time-aligned interaction that enhances TST's performance.

The TST (1:1) configuration outperforms other setups, including multi-token and fusion bottleneck approaches, likely due to its ability to maintain precise, time-aligned interactions between product features. This configuration effectively balances information sharing and efficiency, making it the most optimal choice for multimodal sequential recommendation in our scenario.

### 4.6. Complexity and Runtime Analysis

The time complexity of MTSTRec is primarily dominated by its Transformer layers, each with a standard complexity of $O(n \cdot 2^d)$, where $n$ is the sequence length and $d$ is the embedding dimension. Naively fusing $m$ modalities would result in a worst-case complexity of $O(m \cdot n^2 \cdot d)$. However,

our Time-aligned Shared Token (TST) module introduces only one extra token per time step (plus a prediction token), incurring minimal overhead in practice. With optimized implementations, MTSTRec typically involves only a minor constant overhead compared to standard Transformer-based sequential recommendation models. For completeness, we report parameter sizes, training durations, and inference times for all models in the Appendix M. We further analyze the impact of output token configurations on model performance, presented in Appendix N.

## 5. Conclusion

We introduce MTSTRec, a novel Multimodal Time-aligned Shared Token Recommender that fuses and transmits essential information across different modalities. Our approach allows for precise integration of multimodal features such as product IDs, images, text, and prices while maintaining the unique contributions of each modality. Extensive experimentation shows that MTSTRec significantly outperforms state-of-the-art baselines across various evaluation metrics on real-world e-commerce datasets. Our ablation studies revealed the critical role of each feature type in improving recommendation accuracy, particularly the importance of item identity and textual descriptions in different e-commerce scenarios. Moreover, we showed that the proposed TST fusion method consistently surpasses both early and late fusion strategies by enabling cross-modal interaction that aligns in time throughout the sequence. In summary, MTSTRec represents a significant advancement in multimodal sequential recommendation, offering a flexible and efficient framework that can be adapted to various e-commerce applications. Future work will focus on extending the model to other domains and exploring additional multimodal features for even more personalized recommendations.

## Acknowledgments

This work was supported in part by the National Science and Technology Council (NSTC) of Taiwan under grant number 113-2622-E-002-015. We thank the National Center for High-performance Computing (NCHC) of National Applied Research Laboratories (NARLabs) in Taiwan for providing computational and storage resources. We are also grateful to AviviD.ai for providing the datasets.

## Impact Statement

This research advances sequential recommendation systems with a novel transformer-based framework for efficient multimodal fusion. By aligning temporal sequences across diverse modalities such as text, images, and prices, the approach enhances recommendation accuracy and personalization with broad applications in e-commerce, media, and digital marketing. Its scalable design supports seamless integration of additional modalities, improving user experiences in real-world scenarios. However, we acknowledge potential risks, including possible amplification of biases in multimodal data and privacy concerns regarding data handling practices.

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

## A. Price Extractor

We represent all product prices in the list using a unified normalization approach for consistency. Following the Scalable Numerical Embedding (SCANE) method (Huang et al., 2024), an item price embedding matrix $\boldsymbol{M}^{\text{price}} \in \mathbb{R}^{d \times |I|}$ is constructed, and an input price embedding matrix $\boldsymbol{E}^{\text{price}} \in \mathbb{R}^{d \times n}$, captures the product prices within a user's interaction history. These embeddings are then scaled by their corresponding price value $\boldsymbol{P}_{\boldsymbol{s}_i}$, such that $\boldsymbol{E}_i^{\text{price}} = \boldsymbol{M}_{\boldsymbol{s}_i}^{\text{price}} \times \boldsymbol{P}_{\boldsymbol{s}_i}$ for each item $\boldsymbol{s}_i$ in the sequence, enhancing the model's ability to understand pricing patterns in recommendations.

Furthermore, we experimented with a variant of the SCANE method, where the entire price embedding matrix $\boldsymbol{M}^{\text{price}}$ was replaced such that each element of $\boldsymbol{M}^{\text{price}}$ was set to 1, effectively disabling embedding learning. In this case, the price embeddings were simply the expanded price values $\boldsymbol{P}_{\boldsymbol{s}_i}$ themselves. In our scenario, this variant yielded better performance, as the embedding effectively represented the expanded price without the complexity of learning additional embedding weights.

## B. Implementation Details of the Style Extractor

In our experiments, we extract style features from the first two layers of VGG-19 (Simonyan & Zisserman, 2015) to capture the relevant style information. Each of these layers produces 64 feature maps, resulting in Gram matrices of size $64 \times 64$ (Gatys, 2015; Gatys et al., 2016).

The Gram matrix for each layer is calculated as follows:

Let $\boldsymbol{F}^l \in \mathbb{R}^{Q_l \times R_l}$ represent the feature map from layer $l$, where $Q_l$ represents the number of feature maps (or channels) in the layer, and $R_l$ is the total number of spatial positions, calculated as the height multiplied by the width of the feature map. The Gram matrix $\boldsymbol{G}^l \in \mathbb{R}^{Q_l \times Q_l}$ for layer $l$ is calculated as the inner product of the vectorized feature maps $F_{ik}^l$ and $F_{jk}^l$ at spatial positions $k$, across all feature maps $i$ and $j$. Mathematically, this can be expressed as:

$$\boldsymbol{G}_{ij}^l = \sum_{k=1}^{R_l} \boldsymbol{F}_{ik}^l \boldsymbol{F}_{jk}^l, \tag{9}$$

where $G_{ij}^l$ represents the element of the Gram matrix that captures the correlation between feature map $i$ and feature map $j$ in layer $l$, and the summation over $k$ accounts for all $R_l$ spatial positions in the feature map. After calculating the Gram matrices, we apply max-pooling to compress these matrices, which reduces the computational complexity while retaining the essential style information. This step compresses the Gram matrices to $2 \times 16 \times 16$ style embeddings.

Once each image undergoes the process of Gram matrix computation and subsequent max-pooling, the compressed and flattened Gram matrices for all images are concatenated across layers to form the item style embedding matrix $\boldsymbol{M}^{\text{style}} \in \mathbb{R}^{d \times |I|}$. This matrix encapsulates the style features of all items in the dataset. The input style embedding matrix for each session $\boldsymbol{E}^{\text{style}} \in \mathbb{R}^{d \times n}$ is then retrieved, such that $\boldsymbol{E}_i^{\text{style}} = \boldsymbol{M}_{\boldsymbol{s}_i}^{\text{style}}$, matching item $\boldsymbol{s}_i$ within the user's interaction history.

## C. Comparison of Language Models for Text Embedding

We evaluated the pre-trained language model, BERT (Devlin et al., 2019), and large language models, text-embedding-3-large (OpenAI, 2024a), Llama 3 (Dubey et al., 2024), and Llama 3.1 (Dubey et al., 2024). These comparisons were conducted with the MTSTRec model, focusing solely on product ID and text. The handling of product IDs is discussed in Section 3.2.1. For the text portion, we used product titles and descriptions, converting them into embeddings via the respective Language Models, followed by a linear reduction layer that reduces the embedding dimension to $d$.

The results are presented in Table 5. Llama 3.1 achieves the best performance across all evaluation metrics (NDCG, HR, and MRR). Thus, we choose Llama 3.1 as the backbone for generating text embeddings in our main experiments.

## D. Implementation Details and Results of Prompt-Text Feature

As outlined in Section 3.2.4, we employed various prompting strategies to generate rich textual information, with detailed examples of these prompts provided in Table 6. For Basic Prompt, we utilize three variations: $P_{para}$, $P_{tags}$, and $P_{guess}$. $P_{para}$ prompts the LLM to rephrase the original product description while retaining the same information. $P_{tags}$ aims for a

*Table 5.* Comparison of language models for text embedding in MTSTRec.

| FEATURE | NDCG@5 | HR@5 | MRR@5 |
|---|---|---|---|
| ID | 0.8554 | 0.8074 | 0.7792 |
| ID + TEXT$_{\text{BERT}}$ | 0.8585 | 0.8108 | 0.7829 |
| ID + TEXT$_{\text{OPENAI}}$ | 0.8594 | 0.8119 | 0.7838 |
| ID + TEXT$_{\text{LLAMA3}}$ | 0.8732 | 0.8329 | 0.8000 |
| ID + TEXT$_{\text{LLAMA3.1}}$ | **0.8754** | **0.8340** | **0.8037** |

*Table 6.* All prompts in the prompt-text feature extraction module.

| PROMPT TYPE | SYSTEM AND USER PROMPT |
|---|---|
| $P_{para}$ | **SYSTEM**: *"FFE is an e-commerce website that sells fresh, healthy, high-quality food products without unnecessary additives. You will be provided with the product title and description sold on this e-commerce website. Your task is to paraphrase them."* 
 **USER**: *"Title: `product title`, Description: `product description`."* |
| $P_{tags}$ | **SYSTEM**: *"FFE is an e-commerce website that sells fresh, healthy, high-quality food products without unnecessary additives. You will be provided with the product title and description sold on this e-commerce website. Your task is to summarize this product using tags."* 
 **USER**: *"Title: `product title`, Description: `product description`."* |
| $P_{guess}$ | **SYSTEM**: *"FFE is an e-commerce website that sells fresh, healthy, high-quality food products without unnecessary additives. You will be provided with the product title and description sold on this e-commerce website. Your task is to infer what other products on the site a consumer might be interested in if they purchase this product."* 
 **USER**: *"Title: `product title`, Description: `product description`."* |
| $P_{para}^{rec}$ | **SYSTEM**: *"FFE is an e-commerce website that sells fresh, healthy, high-quality food products without unnecessary additives. Your task is to tell me what else I should say if I want to recommend this product to someone."* 
 **USER**: *"Title: `product title`, Description: `product description`."* |
| $P_{tags}^{rec}$ | **SYSTEM**: *"FFE is an e-commerce website that sells fresh, healthy, high-quality food products without unnecessary additives. Your task is to tell me which tags should be used if I want to recommend this product to someone."* 
 **USER**: *"Title: `product title`, Description: `product description`."* |

concise summary using tags, guiding the LLM to extract key details. Lastly, $P_{guess}$ prompts the LLM to predict what other items the user might purchase based on the product's title and description. The Recommendation Prompt extends the Basic Prompt by introducing a recommendation-oriented task. We define two variations: $P_{para}^{rec}$ and $P_{tags}^{rec}$.

In our experiments, we explored text generation with Llama 3.1 (Dubey et al., 2024) and GPT-4o-mini (Achiam et al., 2023; OpenAI, 2024b) and compared their effectiveness in creating useful text embeddings. Each prompt was carefully designed with specific settings to match its intended function.

For instance:

- $P_{para}$: This prompt was used to paraphrase the product title and description, configured with a temperature of 0.7, a maximum token limit of 256, and top_p = 1.

- $P_{tags}$: For this prompt, which focuses on summarizing product information using tags, we set the temperature to 0.5, a maximum token limit of 128, and top_p = 1.

- $P_{guess}$: This prompt aims to infer potential additional products the user might purchase based on the current product. It was configured with a temperature of 1, a maximum token limit of 512, and top_p = 1.

- $P_{para}^{rec}$: This recommendation-focused paraphrase prompt had a temperature of 1, a maximum token limit of 384, and top_p = 1.

- $P_{tags}^{rec}$: Designed to summarize product information for recommendations, this prompt used a temperature of 1, a maximum token limit of 128, and top_p = 1.

The text generated by Llama 3.1 (Dubey et al., 2024) or GPT-4o-mini (Achiam et al., 2023; OpenAI, 2024b) was uniformly converted into embeddings using Llama 3.1 (shown in Appendix C) to maintain consistency across experiments. This approach allowed for reliable comparisons, focusing on how different prompts and their corresponding embeddings affected recommendation accuracy in MTSTRec. Each experiment incorporated prompt embeddings alongside the product ID to measure performance.

We also assessed the influence of different LLMs (Llama 3.1 and GPT-4o-mini) on overall model performance by comparing how each model's generated text, once converted into embeddings, impacted recommendation results. The detailed outcomes are summarized in the following Table 7.

*Table 7.* Comparison of prompt strategies and LLMs (Llama 3.1 vs. GPT-4o-mini) in MTSTRec.

| FEATURE | PROMPT STRATEGY | LLAMA 3.1 | GPT-4O-MINI |
|---------|-----------------|-----------|-------------|
| ID + PROMPT | $P_{para}$ | 0.8727 | 0.8747 |
| | $P_{tags}$ | 0.8727 | 0.8723 |
| | $P_{guess}$ | 0.8718 | 0.8702 |
| | $P_{para}^{rec}$ | 0.8697 | 0.8703 |
| | $P_{tags}^{rec}$ | 0.8725 | 0.8716 |
| | **AVERAGE** | 0.8719 | 0.8718 |

Based on the results, we observe that incorporating prompt embeddings into the model helps improve performance compared to using only the product ID. While the results for prompt embeddings are not as high as directly using text embeddings, this might suggest that the original product titles and descriptions, when processed by the LLM, sufficiently capture the characteristics of the items. However, our findings demonstrate that prompt embeddings positively influence recommendation outcomes. Additionally, the prompt embeddings generated by both Llama 3.1 and GPT-4o-mini show comparable performance across the five strategies. Llama 3.1 has a slight edge on average, which is why we chose to primarily use Llama 3.1 to generate our prompt-text features in subsequent experiments.

## E. Gating Weights and Performance of Prompt-Text Feature

As explained in Section 3.2.4, a gating network is applied to the five prompt strategies to learn their relative importance. The gating weights (Table 8) are derived from the validation results of the ID + Text + Prompt features. In addition, we compare the performance of ID + Text versus ID + Text + Prompt, demonstrating improved recommendation accuracy with the inclusion of prompt embeddings, regardless of the LLM used, as shown in Table 9.

*Table 8.* Gating weights for different prompt strategies.

| PROMPT STRATEGY | LLAMA 3.1 | GPT-4O-MINI |
|-----------------|-----------|-------------|
| $P_{para}$ | 0.1391 | **0.2766** |
| $P_{tags}$ | **0.4485** | 0.1846 |
| $P_{guess}$ | 0.1367 | 0.1403 |
| $P_{para}^{rec}$ | 0.1328 | 0.1325 |
| $P_{tags}^{rec}$ | 0.1428 | 0.2659 |

*Table 9.* Performance comparison of ID + text and ID + text + prompt-text in MTSTRec.

| FEATURE | NDCG@5 |
|---------|--------|
| ID + TEXT | 0.8754 |
| ID + TEXT + PROMPT (GPT-4O-MINI) | 0.8790 |
| ID + TEXT + PROMPT (LLAMA 3.1) | **0.8795** |

In conclusion, adding prompt embeddings significantly enhances recommendation performance, as shown by the improved NDCG@5 scores. Furthermore, based on the Llama 3.1 gating weights, $P_{tags}$ emerges as a critical factor in improving recommendation accuracy. Similarly, the gating weights from GPT-4o-mini align with the results from individual prompt embeddings, reinforcing the importance of specific prompts, such as $P_{para}$, $P_{tags}$, and $P_{tags}^{rec}$, in optimizing model performance shown in Table 7.

## F. Detailed Architecture and Process of the Self-Attention Encoder

Self-Attention Encoder architecture follows the transformer encoder architecture proposed by Vaswani (2017). The input embedding from each modality $\boldsymbol{Z}$ is processed independently through an independent transformer encoder with multiple layers. Each layer consists of three key components: Multi-Head Self-Attention (MSA), Layer Normalization (LN), and Multilayer Perceptron (MLP), all connected via residual connections. The transformation at layer $l$ is defined as follows:

$$\boldsymbol{Y}^{l-1} = \text{MSA}(\text{LN}(\boldsymbol{Z}^{l-1})) + \text{LN}(\boldsymbol{Z}^{l-1}), \tag{10}$$

$$\boldsymbol{Z}^l = \text{MLP}(\text{LN}(\boldsymbol{Y}^{l-1})). \tag{11}$$

In the MSA block, self-attention computes the attention scores between tokens, allowing each token to dynamically attend to other tokens in the sequence. The attention mechanism is defined as:

$$\text{MSA}(\mathbf{X}) = \text{Attention}(\mathbf{W}^Q\mathbf{X}, \mathbf{W}^K\mathbf{X}, \mathbf{W}^V\mathbf{X}), \tag{12}$$

where $\mathbf{W}^Q$, $\mathbf{W}^K$, and $\mathbf{W}^V$ are the weight matrices used to transform the input tensor $\mathbf{X}$ into queries, keys, and values, respectively. The attention scores are calculated as the dot product between queries and keys, enabling the model to prioritize the most important parts of the input.

## G. Dataset

Our experiments utilize three datasets: *Food E-commerce*, *House-Hold E-commerce*, *H&M (Trousers)*. The *Food E-commerce* dataset includes a total of 1,507,388 interactions, with an average session length of 4.693 and a total of 216,576 sessions. The *House-Hold E-commerce* dataset consists of 94,984 interactions, an average session length of 5.914, and a total of 12,345 sessions. The *H&M (Trousers)* dataset contains user 3,576,972 purchase records and a total of 416,794 sessions, with further details provided in Table 10.

*Table 10.* Statistics of datasets.

| DATASET | Food | House-Hold | H&M (Trousers) |
|---|---|---|---|
| # SESSIONS | 216,576 | 12,345 | 416,794 |
| # PRODUCTS | 770 | 2,464 | 11,150 |
| AVG. SESSION | 4.693 | 5.914 | 7.029 |
| AVG. PURCHASE | 2.267 | 1.780 | 1.553 |
| # ACTIONS | 1,507,388 | 94,984 | 3,576,972 |

We apply preprocessing steps to filter out sessions with fewer than three interactions to ensure sufficient data for model training. Session lengths are limited to 20 items. For shorter sessions, we pad them with zeros, while for longer sessions, we retain only the last 20 products, removing any repeat items at the end of the sequence. This adjustment allows the model to better focus on complex patterns, enhancing its applicability to real-world scenarios.

For the first two datasets, each session's purchase order involves multiple items, so we treat the answer set as a multiple-choice task. The same preprocessing steps are applied to the *H&M (Trousers)* dataset. Although it consists solely of purchase actions, we sort the items by purchase time and treat the products bought on the last day as the answer set. Since users may purchase multiple items on the last day, this is also treated as a multiple-choice task. To facilitate model computation, we pad all answers to a fixed length of 50 items.

**Dataset Splitting.** Each session in the proprietary datasets consists of historical clicks and a final purchase order. The data is split chronologically into 75% for training, 12.5% for validation, and 12.5% for testing based on the purchase orders. For the *H&M (Trousers)* dataset, which contains only purchase actions, items are sorted by purchase time, and those bought on the last day are used as the answer set, ensuring consistency across all datasets (Meng et al., 2020). The model predicts items in purchase orders as a multi-label recommendation task, where each sequence may have multiple correct answers (purchases).

## H. Evaluation Metrics

For each session, we randomly sampled 100 items that the user did not interact with under the target behavior as negative samples. Finally, we report the results on the test set while selecting the best hyperparameters using the validation set. We detail the three evaluation metrics used in our experiments: NDCG@k, HR@k, and MRR@k, where k is set to 5 and 10. Since our problem involves multiple correct answers (multi-label), we have adjusted the definitions of HR@k and MRR@k accordingly. HR is calculated by treating each correct answer as a separate single-choice task. The model ranks each correct item among negative samples, and we calculate the hit rate for each task. The final HR@k is the average of these hit rates across all correct answers in the sequence. MRR@k follows a similar approach, where we compute the reciprocal rank for each correct answer, and the final MRR@k is the average across all correct answers. Below are the definitions of each metric, along with an example calculation.

**NDCG@k (Normalized Discounted Cumulative Gain).** NDCG@K considers both the relevance and the position of the correct items in the ranked list, with higher-ranked relevant items contributing more to the score.

The Discounted Cumulative Gain (DCG@k) is calculated by summing the relevance scores of the correct items, where the relevance score decreases logarithmically based on the item's rank position. The formula for DCG@k is:

$$\text{DCG@k} = \sum_{r=1}^{k} \frac{rel(r)}{\log_2(r+1)}, \tag{13}$$

where $rel(r)$ represents the relevance score of the item at rank $r$, where all items in the purchase order are assigned a relevance score of 1. Higher-ranked relevant items contribute more to the final score, as their positions are weighted more heavily in the DCG calculation.

The Ideal DCG (IDCG@k) represents the best possible ranking scenario, where all relevant items are ranked at the top. Since we are normalizing the DCG score, the ideal ranking is computed by assuming the best-case relevance distribution. The formula for IDCG@k is:

$$\text{IDCG@k} = \sum_{r=1}^{\min(k,n)} \frac{1}{\log_2(r+1)}, \tag{14}$$

where $\min(k, n)$ ensures that we only consider the smaller of $k$ (the cutoff) and $n$ (the number of correct answers).

Finally, the NDCG@k is calculated by normalizing DCG by the ideal DCG. The formula for NDCG@k is:

$$\text{NDCG@k} = \frac{\text{DCG@k}}{\text{IDCG@k}}. \tag{15}$$

This normalization makes NDCG range between 0 and 1, allowing consistent comparison across different queries. A higher NDCG score indicates that the correct items are ranked closer to the top.

**HR@k (Hit Rate).** HR@k is calculated by checking whether any of the ground truth items appear within the top k-ranked items. For the multi-label task, we calculate HR for each correct item and then average the results. The formula for HR@k is:

$$\text{HR@k} = \frac{1}{|n|} \sum_{i=1}^{|n|} \mathbf{1}[\text{rank}_i \leq k], \tag{16}$$

where $\mathbf{1}[\text{rank}_i \leq k]$ is 1 if the correct item $i$ is ranked within the top k, and 0 otherwise.

**MRR@k (Mean Reciprocal Rank).** MRR@k measures the ranking of the first correct item within the top k positions. For the multi-label task, MRR@k is calculated by averaging the reciprocal ranks of the correct items:

$$\text{MRR@k} = \frac{1}{|\text{n}|} \sum_{i=1}^{|n|} \frac{1}{rank_i}, \tag{17}$$

where $rank_i$ is the rank position of the correct item $i$.

## I. Benchmark Models

- **SASRec** (Kang & McAuley, 2018): A self-attention-based sequential model with causal masking that captures long-term user preferences by attending only to previous tokens.

- **BERT4Rec** (Sun et al., 2019): A bidirectional sequential recommendation model that uses self-attention to predict masked items in user behavior sequences, capturing both left and right context.

- **SASRec$^{+}$**: An enhanced version of SASRec (Kang & McAuley, 2018), which integrates item ID, text, image features. The text and image features are processed using the techniques from our MTSTRec model. To ensure stable training, we reduce the dimensionality to 256, as the model would otherwise fail to converge properly.

- **BERT4Rec$^{+}$**: An enhanced version of BERT4Rec (Sun et al., 2019), which integrates item ID, text, image features. The text and image features are processed using the techniques from our MTSTRec model. To ensure stable training, we reduce the dimensionality to 256, as the model would otherwise fail to converge properly.

- **MMMLP** (Liang et al., 2023): A multimodal MLP-based model that processes text, image, and price features (with price added for fair comparison) through a Feature Mixer Layer, Fusion Mixer Layer, and Prediction Layer, achieving state-of-the-art performance with linear complexity.

- **MMMLP$^+$**: An enhanced version of MMMLP (Liang et al., 2023), which integrates item ID, image, text, prompt-text, and price features. The features are processed using the same techniques from our MTSTRec model, ensuring that all five feature embeddings extracted from the feature extractor are identical. This ensures a fair comparison that focuses solely on the differences in the predictive model architectures.

## J. Implementation details

In our experiments, we tuned the hyperparameters based on validation data to ensure optimal performance. The batch size was uniformly set to 64 for all models, and the input dimension $d$ was fixed at 512. We employed the AdamW optimizer while the maximum sequence length $N$ was set to 20. The fusion layers were standardized across models, with $L_{fusion} = 3$ and a dropout rate of 0.1.

For the ID feature encoder, we used two transformer blocks ($L_{id} = 2$) with four attention heads, applying a hidden layer dropout of 0.1 to maintain fairness with other benchmark models.

For other feature encoders, such as text, image, and price, we experimented with different settings. The number of each encoder layer ($L_{mod}$) was tested across values of $\{2, 4, 8\}$, and the number of attention heads across $\{1, 2, 4, 8, 16\}$. We also experimented with dropout rates of $\{0.1, 0.2, 0.3\}$ in the hidden layers. The learning rate was tested across a range of $\{0.001, 0.0005, 0.0001, 0.00005, 0.00001\}$, while the L2 regularization penalty was tuned from $\{0.0001, 0.00005, 0.00001, 0.000005, 0.000001\}$. A gamma value of $\{0.9, 0.75, 0.5\}$ was set for learning rate decay.

For the baseline models (e.g., SASRec (Kang & McAuley, 2018), BERT4Rec (Sun et al., 2019)), we ensured that key settings such as batch size, the number of encoder blocks, and attention heads were aligned with our model for fair comparison. However, for other settings, we followed the recommended configurations in the original papers.

## K. The Impact of ID Modules Across Different Datasets

This section examines the performance impact of removing ID modules in the MTSTRec model across three datasets: *Food E-commerce*, *House-Hold E-commerce*, and *H&M (Trousers)*. Table 11 highlights notable performance differences between the full model and the version without ID modules. The results show that ID modules play a crucial role in *Food E-commerce* and *House-Hold E-commerce* datasets, significantly boosting recommendation accuracy. In contrast, their removal in the *H&M (Trousers)* dataset unexpectedly improves performance.

*Table 11.* The impact of removing ID modules across e-commerce platforms.

| ABLATION STUDY | Food E-commerce | | | House-Hold E-commerce | | | H&M (Trousers) | | |
|---|---|---|---|---|---|---|---|---|---|
| | NDCG@5 | HR@5 | MRR@5 | NDCG@5 | HR@5 | MRR@5 | NDCG@5 | HR@5 | MRR@5 |
| MTSTREC | **0.8800** | **0.8407** | **0.8086** | **0.8942** | **0.9067** | **0.8568** | 0.2307 | 0.3139 | 0.1871 |
| W/O ID | 0.7582 | 0.7506 | 0.6459 | 0.7913 | 0.8337 | 0.7184 | **0.2724** | **0.3676** | **0.2232** |

In the *Food E-commerce* and *House-Hold E-commerce* datasets, the removal of ID modules results in a noticeable decline in all metrics. For example, in the *Food E-commerce* dataset, NDCG@5 drops from 0.8800 to 0.7582, while HR@5 decreases from 0.8407 to 0.7506. Similarly, in the *House-Hold E-commerce* dataset, NDCG@5 decreases from 0.8942 to 0.7913, with HR@5 following a similar trend. One primary reason for this decline is the smaller number of unique products in these datasets, as shown in Table 10. With only 770 and 2,464 products, respectively, these datasets allow product IDs to serve as highly discriminative features, enabling the model to learn precise user-item interactions. The supervised learning framework further reinforces this dependency, as the model optimizes embeddings for individual IDs, which are particularly effective in smaller product spaces.

Conversely, in the *H&M (Trousers)* dataset, the removal of ID modules results in a performance improvement, with NDCG@5 increasing from 0.2307 to 0.2724 and similar gains observed in HR@5 and MRR@5. This improvement can be attributed to the larger and more diverse product space, which includes 11,150 unique items. The increased number of products introduces sparsity in user-product interactions, making ID embeddings less effective at representing individual

items. Moreover, the *H&M (Trousers)* dataset defines items with identical styles but different colors as separate products, which complicates the embedding space and reduces the distinctiveness of product IDs.

These findings underscore the dataset-dependent role of ID modules. Product IDs are critical for achieving high recommendation accuracy in domains with fewer products and consistent user behavior. However, in datasets with larger product spaces and greater complexity, the reliance on IDs diminishes, requiring models to leverage other features. This highlights the importance of multimodality, as incorporating information from other modalities (e.g., images or text) provides richer insights, enabling models to adapt better to diverse datasets.

## L. Configuration Details of the Shared Token

- **Time-aligned Shared Tokens (TST) (1:1)**, used in our MTSTRec model, pairs each time step in the sequence with a corresponding shared token in a 1:1 relationship. This means that each product in the sequence is aligned with a single shared token, allowing features from different modalities of the same product to interact and share information at that specific time step. This design ensures that information sharing is precise and time-aligned, leading to more accurate feature fusion.

- **TST (1:2) & TST (1:4)** include configurations where each time step is associated with multiple shared tokens rather than just one. For example, in **TST (1:2)**, each product in the sequence is paired with two shared tokens, and in **TST (1:4)**, each product is paired with four shared tokens. These variants allow for more extensive information sharing at each time step.

- **Fusion Bottlenecks (all:4+1)** is based on a configuration from Google (Nagrani et al., 2021) originally designed for sequence fusion classification tasks involving image and speech data. In this setting, shared tokens are not tied to a specific time step but attend to the entire sequence, enabling broader information exchange across the sequence. For a fair comparison in our multimodal sequential recommendation task, we adapted this method to a **Fusion Bottlenecks (all:20+1)** configuration, matching the number of tokens used in our TST approach.

## M. Parameter Size and Runtime Analysis

We report detailed statistics on the number of parameters, training time, and inference time for all models evaluated on the *House-Hold E-commerce* dataset. As shown in Table 12, MTSTRec incurs a longer training time due to its multimodal architecture and the inclusion of the Time-aligned Shared Token (TST) module. Nevertheless, its inference time remains competitive, only marginally higher than that of other multimodal baselines. Despite incorporating additional modality-specific encoders and fusion layers, MTSTRec is significantly more compact than multimodal baselines such as SASRec$^+$, BERT4Rec$^+$, MMMLP, and MMMLP$^+$. These results indicate that MTSTRec achieves a favorable balance between model size and effectiveness without compromising expressiveness or accuracy.

*Table 12.* Parameter size, training time, and inference time on the *House-Hold E-commerce* dataset.

| MODEL | PARAMETER SIZE (MILLION) | TRAINING TIME (MINUTE) | INFERENCE TIME (SECOND) |
|---|---|---|---|
| SASREC | 4.69 | 4.5 | 1.86 |
| BERT4REC | 7.84 | 4.1 | 1.88 |
| SASREC$^+$ | 203.67 | 66 | 7.48 |
| BERT4REC$^+$ | 400.33 | 59 | 7.89 |
| MMMLP | 97.38 | 33 | 6.56 |
| MMMLP$^+$ | 174.54 | 39 | 7.27 |
| **MTSTREC** | 59.24 | 83 | 12.43 |

## N. Output Token Design Choices

### N.1. Use of Modality-Specific CLOZE Tokens ($z_{cz}$)

During model development, we compared two strategies for summarizing each modality's sequence: (i) using the output of the last token, and (ii) appending a dedicated CLOZE token ($z_{cz}$) at the end of each modality sequence. As shown in Table 13, the latter consistently outperformed the former on the *House-Hold E-commerce* dataset. Introducing a $z_{cz}$ token

per modality provides a dedicated placeholder for the next-item prediction, enabling the model to aggregate relevant signals from each modality's perspective. This led to clearer and more informative multimodal representations for downstream prediction.

*Table 13.* Performance impact of modality-specific CLOZE token ($z_{cz}$) on the *House-Hold E-commerce* dataset.

| MODEL | NDCG@5 | HR@5 | MRR@5 |
|---|---|---|---|
| MTSTREC | **0.8942** (±0.0035) | **0.9067** (±0.0024) | **0.8568** (±0.0041) |
| W/O CLOZE | 0.8115 (±0.0023) | 0.8275 (±0.0027) | 0.7553 (±0.0043) |

### N.2. Excluding the Shared Token ($z_{sh}$) from Final Output

During model development, we also investigated whether including the shared fusion token $z_{sh}$ in the final prediction output would improve model performance. While $z_{sh}$ plays a crucial role in aligning cross-modal features during the fusion, our experiments showed that naively incorporating it into the output layer degraded prediction accuracy. The comparative results are summarized in Table 14.

*Table 14.* Performance impact of the shared token ($z_{sh}$) in the final output on the *House-Hold E-commerce* dataset.

| MODEL | NDCG@5 | HR@5 | MRR@5 |
|---|---|---|---|
| MTSTREC | **0.8942** (±0.0035) | **0.9067** (±0.0024) | **0.8568** (±0.0041) |
| CONCATENATE $z_{sh}$ | 0.8106 (±0.0089) | 0.8692 (±0.0067) | 0.7142 (±0.0108) |

## O. Dataset Release Document

### O.1. Introduction

We have released two datasets, *Food E-commerce* and *House-Hold E-commerce*, both collected with full user consent from e-commerce platforms. These datasets contain user interaction data from October 2023 to June 2024, including page views and purchases.

### O.2. Dataset Overview

The *Food E-commerce* and *House-Hold E-commerce* offer a rich source of user interaction data. Since each order can contain multiple purchased products, each sequence may have multiple correct answers (purchases). We provide a statistical summary of the raw data in Table 10. The statistics include key metrics such as `#Sessions`, representing the number of user sessions in the dataset; `#Products`, indicating the total number of unique products; `Avg. Session`, which is the average length of user sessions; `Avg. Purchase`, the average number of products purchased per order; and `#Actions`, capturing the total count of product browsing records and purchase actions.

### O.3. Feature Preprocessing

**Image Style Embedding:** The style features of product images are extracted using the first two layers of the VGG-19 (Simonyan & Zisserman, 2015) network. We calculate Gram matrices (Gatys, 2015; Gatys et al., 2016) from these layers to capture the relationships between feature maps and then apply max-pooling to compress the style information. Each image is ultimately represented as a 512-dimensional embedding that summarizes its style characteristics.

**Text Embedding:** For textual features, we process product titles and descriptions using Llama 3.1 (Dubey et al., 2024). This generates a 4096-dimensional embedding for each product. Additionally, we create five prompt-based text embeddings to highlight different aspects of the product information. These embeddings provide a comprehensive representation of the textual data. (see more details in Section 3.2.4)

### O.4. File Location

The files containing the datasets and related resources are organized in a structured directory format for easy access and navigation. Below is an overview of the file locations, using the *Food E-commerce* dataset as an example. The *House-Hold*

*E-commerce* dataset follows the same directory structure.

- **Food/browse_seq_train, browse_seq_val, browse_seq_test:** These files contain the training, validation, and testing sequences of user interactions, including browsing records and purchase actions within *Food E-commerce*.

- **Food/sessiondate_train, sessiondate_val, sessiondate_test:** These files contain the purchase date of training, validation, and testing data within *Food E-commerce*. The corresponding purchase date for each session is provided to determine which recommended products can be offered.

- **Food/Product_Feature/PriceFeature:** This file contains the sale price feature of each product within *Food E-commerce*.

- **Food/Product_Feature/Image_Style_Embedding:** This file contains the style embeddings of each product within *Food E-commerce* .

- **Food/Product_Feature/Text_Embedding_TitleDescription, Text_Embedding_Basic_Paraphrase, Text_Embedding_Basic_Tags, Text_Embedding_Basic_Guess, Text_Embedding_Rec_Paraphrase, Text_Embedding_Rec_Tags:** These files contain the different text embeddings of each product within *Food E-commerce* including Title and Description, Basic Paraphrase Prompt, Basic Tags Prompt, Basic Guess Prompt, Recommendation Paraphrase Prompt, and Recommendation Tags Prompt.

- **Food/Month_Product/2401_productlist:** This file contains the product index for the *Food E-commerce* platform in January 2024. It provides a snapshot of the products available at the beginning of the month, serving as a reference for predicting which items should be recommended based on their corresponding purchase dates. The folder Month_Product contains eight files, namely: 2310_productlist, 2311_productlist, 2312_productlist, 2401_productlist, 2402_productlist, 2403_productlist, 2404_productlist, and 2406_productlist.

### O.5. Dataset Example

The following examples illustrate the structure and content of the datasets used in this research. Each entry in the dataset consists of various encoded features that represent different aspects of user interactions, item properties, and contextual information. The examples below showcase typical data points, including encoded identifiers, interaction sequences, and embeddings extracted from pre-trained language models. These examples are representative of the format and type of data that researchers will work with when utilizing *Food E-commerce* and *House-Hold E-commerce* datasets for their studies.

**browse_seq**
Content format (each line): `[[browse_product_idx],[purchase_product_idx]]` (The product IDs start from 1.)
Example: `[[1, 57, 35, 102, 435], [720, 102, 35]]`

**sessiondate**
Content format (each line): `purchase_datetime`
Example: `datetime.datetime(2024, 3, 16, 0, 0, 0, 0)`

**PriceFeature**
Content format: `every product sale price (normalization)` (From the price of the product with ID 1 to the product with ID $n + 1$.)
Example: `[0.5035971223021583, 0.5392953929539296, ..., 0.6254901960784314, 0.6235294117647059]`

**Image_Style_Embedding**
Content format (each line): `gram matrixes of a product (dim=512)` (The lines are sorted based on the product index. Since our index starts from 1, the first line is filled with an all-zero embedding.)
Example: `tensor([0.0323, 0.0204, ..., 0.0097, 0.0167])`

**Text_Embedding**
Content format (each line): `Llama 3.1 embedding of a product (dim=4096)` (The lines are sorted based on the product index. Since our index starts from 1, the first line is filled with an all-zero embedding.)
Example: `tensor([1.7661, -0.1427, ..., -0.3347, -0.9397])`

**2403_productlist**

Content format: `product idx list`

Example: `[587, 588, ..., 183, 184]`

### O.6. Conclusion

This technical appendix provides a comprehensive overview of the datasets released as part of our research. The *Food E-commerce* and *House-Hold E-commerce* datasets serve as valuable resources for advancing the study of multimodal recommendation systems. With their diverse interaction data and carefully curated features, these datasets offer a unique opportunity for researchers to develop and evaluate cutting-edge models. The provided file locations and preprocessing steps ensure that users can easily access and utilize these datasets for further research and development efforts.

