# OpenReview forum: "MTSTRec: Multimodal Time-Aligned Shared Token Recommender"
_ICML.cc/2025/Conference — ICML 2025 poster_

### Official Review · Reviewer_AnDS · 2025-03-13

**Overall Recommendation:** 2

**Summary:**

The paper introduces MTSTRec, a transformer-based multimodal recommendation framework that temporally aligns different modalities to improve sequential recommendations. Unlike existing methods, Unlike existing methods that perform either early or late fusion, MTSTRec employs a Time-aligned Shared Token (TST) module for intermediate fusion, ensuring better cross-modal alignment while preserving the unique contributions of each modality. Extensive experiments on multiple datasets demonstrate its superiority over baseline models, with ablation studies highlighting the importance of different modalities and the effectiveness of the TST module.

## update after rebuttal
Since the authors did not add new results I mentioned in my comments, I will keep the score unchanged.

**Claims And Evidence:**

The authors provide evidence for the effectiveness of the proposed time-aligned fusion module through ablation studies on multiple datasets. However, two key issues remain:
1.The discussion on multimodal fusion strategies is limited, particularly regarding mid-fusion, which is central to this work. A more thorough comparison with existing approaches would strengthen the contribution.
2.The experiments lack baselines from multimodal sequential recommendation models, making it difficult to fully assess the model’s novelty and effectiveness. Including such baselines would provide a clearer evaluation of its improvements.

**Essential References Not Discussed:**

The paper lacks a sufficiently comprehensive discussion of relevant literature on multimodal fusion strategies, including but not limited to the following references, which were either not discussed or mentioned without being compared as baselines:
[1] Zhou, Xin, and Zhiqi Shen. "A tale of two graphs: Freezing and denoising graph structures for multimodal recommendation." Proceedings of the 31st ACM International Conference on Multimedia. 2023.
[2] Zhong, Shanshan, et al. "Mirror Gradient: Towards Robust Multimodal Recommender Systems via Exploring Flat Local Minima." Proceedings of the ACM on Web Conference 2024. 2024.
[3] Jiang, Hao, et al. "What aspect do you like: Multi-scale time-aware user interest modeling for micro-video recommendation." Proceedings of the 28th ACM International conference on Multimedia. 2020.

**Experimental Designs Or Analyses:**

1.Multiple datasets are used for evaluation, including one public dataset and two new datasets.
2.The detailed ablation studies effectively demonstrate the impact of individual features and fusion mechanisms.
3.The paper lacks a thorough comparison with more state-of-the-art multimodal recommendation methods, limiting the context for evaluating the model's novelty.

**Methods And Evaluation Criteria:**

Methods: The TST module effectively integrates multimodal features while maintaining temporal consistency, and extensive ablation studies validate the impact of individual components.
Evaluation Criteria: Multiple datasets are used for evaluation, including one public dataset and two new datasets. However, the three types of multimodal fusion mentioned by the authors should be represented by corresponding baselines.

**Other Comments Or Suggestions:**

1.The writing is clear and flows smoothly, making the paper easy to follow for readers.
2.The discussion of related work is insufficient, and the experimental comparisons are lacking. More attention should be given to the discussion of multimodal fusion techniques.

**Other Strengths And Weaknesses:**

Strengths
1.The time-aligned multimodal fusion module proposed in this paper is highly innovative, and experimental results demonstrate the effectiveness of the token.
2.The paper provides a valuable analysis of multimodal sequential recommendation, offering insights that inspire further research into the role of multimodal fusion at different stages.
Weaknesses
1.The discussion of related work is insufficient, with a lack of thorough literature review.
2.Additionally, the experimental comparisons with similar models are inadequate, leaving the innovation and effectiveness of the approach open to further validation.

**Questions For Authors:**

1.How significant is the contribution of large models in extracting text information to the overall performance of the proposed model?
2.How does the performance of other LLM-based multimodal sequential recommendation models compare to your model?

**Relation To Broader Scientific Literature:**

1.Multimodal Fusion: It proposes a time-aligned fusion module, improving the integration of various modalities in recommendation systems, which advances multimodal learning research.
2.LLM for Feature Extraction and Processing: The work demonstrates the use of large models for extracting text information and handling different feature types, contributing to more effective feature processing in recommendation systems.

**Theoretical Claims:**

The paper provides detailed formulations of the model and the calculation methods for evaluation metrics in both the main text and supplementary materials. However, the theoretical analysis and proof are somewhat limited, with a primary focus on experimental validation.

---

> ### Author Rebuttal · Authors · 2025-04-01
>
> We sincerely thank the reviewer for the thoughtful and constructive feedback. We are especially grateful for the recognition of the strengths of our work, particularly the innovation of the time-aligned multimodal fusion module and its effectiveness demonstrated through our experiments. We also appreciate your kind comments on the clarity and structure of the writing, as well as the usefulness of the supplementary materials.
>
> Below, we address each of your comments and suggestions in detail.
>
>
> **[W1]** Insufficient discussion of related work [Weaknesses] [Experimental Designs Or Analyses 3]
>
> **[R1]** Thank you for your helpful suggestion. The mentioned works ([1], [2], [3]) were included in our original related work section, but due to space limitations, we were unable to discuss them in depth. In the revised version, we will enhance this section by more clearly comparing these methods with ours and highlighting the unique aspects of MTSTRec, particularly its time-aligned mid-fusion design. We appreciate your feedback and will further elaborate on these references if space allows in future revisions.
>
>
> **[W2]** lack of multimodal sequential baselines [Weaknesses] [Claims And Evidence 2] [Essential References Not Discussed]
>
> **[R2]** We respectfully contend that the current experimental design adequately supports the evaluation of MTSTRec’s novelty and effectiveness within the defined scope. In Section 4.1 (Experimental Settings), we included baselines such as SASRec and BERT4Rec (single-modal sequential models), enhanced versions SASRec+ and BERT4Rec+ (early fusion with multimodal features), and MMMLP (a state-of-the-art late-fusion multimodal model). Additionally, MMMLP+ incorporates the same multimodal features as MTSTRec, ensuring a fair comparison across fusion strategies. These models, tested on three diverse datasets (Section 4.2), provide a solid benchmark for assessing MTSTRec’s improvements, as evidenced by its superior performance (e.g., NDCG@5 gains of 3.4%-43.7% over baselines).
>
>
> **[W3]** Insufficient discussion and comparison of multimodal fusion strategies, especially mid-fusion [Claims And Evidence 1] [Methods And Evaluation Criteria] [Other Comments Or Suggestions 2]
>
> **[R3]** Thank you for your feedback. We believe the current content adequately supports MTSTRec’s contributions. Section 2.2 outlines early (VBPR), mid (MM-Rec), and late (MMMLP) fusion approaches, aligning with the three fusion types. Our focus is on the novel Time-aligned Shared Token (TST) module, which distinguishes our mid-fusion by temporally aligning modalities—unlike prior mid-fusion works like MM-Rec. Additionally, Table 4 compares various mid-fusion methods (e.g., TST (1:1), TST (1:2), TST (1:4), and bottlenecks), demonstrating TST’s superiority.
>
>
> **[Q1]** Impact of large language models on textual feature extraction and overall performance [Questions]
>
> **[R4]** Referring to Table 5, larger language models (LLMs) consistently outperform smaller text encoders by capturing more nuanced context, boosting scores on HR@5 and NDCG@5. Despite higher computational overhead, their richer embeddings better align user preferences with item attributes, closing a notable performance gap. (Llama 3.1 excelling in MTSTRec’s text extraction, with NDCG@5: 0.8754, HR@5: 0.8340, outperforming BERT (NDCG@5: 0.8585, HR@5: 0.108) and others. Its richer embeddings improve preference alignment by ~1.97% in NDCG@5)
>
>
> **[Q2]** Comparison with other LLM-based multimodal sequential recommendation models [Questions]
>
> **[R5]** We acknowledge recent progress in LLM-based recommenders such as LLM-Rec. While these approaches inspire our work, MTSTRec is designed for a multimodal sequential recommendation, integrating not only text but also images, prices, and item IDs via our proposed TST module. In contrast, LLM-Rec focuses on text-only scenarios with different modeling objectives. It is worth noting that LLMRec can be seen as analogous to our prompt encoder. Therefore, our ablation studies—comparing with early fusion, late fusion, and using only the prompt encoder—can be viewed as an indirect comparison with LLMRec.

---

### Official Review · Reviewer_2WPs · 2025-03-14

**Overall Recommendation:** 3

**Summary:**

This paper proposes a unified multimodal recommendation framework with a Temporally-aligned Shared Token (TST) fusion module to learn cross-modal interactions, ensuring time-consistent alignment and modality fusion. Comprehensive experiments are conducted to compare the framework with existing works and to validate the effectiveness of different modules.

**Claims And Evidence:**

The authors claim that this work is a unified solution for multi-modal recommendation. However, the evidence presented in Table 2 appears to partially undermine this claim. It can be observed that removing the ID embedding leads to the most significant performance degradation while removing other modalities (e.g., style) results in nearly identical performances. For instance, on the Fresh-Food E-commerce dataset, the NDCG@5 for MTSTRec (0.8800±0.0023) and for MTSTRec_W/O_style (0.8784±0.0020) show negligible difference, suggesting that the multi-modal features may not contribute substantially to the model’s overall performance. The above observations raise significant doubts regarding the model’s effectiveness in multi-modal recommendation scenarios. The minimal performance impact observed when removing specific modalities (e.g., style) suggests that the model may not be fully leveraging the potential of multi-modal features. Instead, the heavy reliance on ID embeddings indicates that the model’s success is predominantly driven by single-modal (ID-based) information. I expect that the authors further clarify the main contributions.

**Essential References Not Discussed:**

N/A. The authors have provided a comprehensive literature review in the related work section.

**Experimental Designs Or Analyses:**

I have carefully checked the experimental designs, results, and analyses, and have two major concerns:
1)	The datasets (Fresh-Food E-Commerce, House-Hold E-commerce, and H&M) are all E-Commerce datasets, which is limited in evaluating the model’s generalizability. Datasets in other domains (e.g., MovieLens, Last-FM, Yelp, etc.) should be considered for evaluation. It would greatly strengthen the paper if the above diverse datasets are used for evaluation.
2)	The ablation studies are all conducted on the Fresh-Food E-commerce, which further limits the method’s generalizability. It remains unclear whether the observed performance improvements and the relative importance of each module hold true for the other two datasets.

**Methods And Evaluation Criteria:**

The proposed method is built upon Transformer-like architectures with a TST fusion module. Since transformer has been applied in both academic research and industrial applications within the domain of Recommender Systems (RS), the soundness of the approach is well-established. For criteria, the authors use the commonly adopted metrics (HitRate, NDCG, and MRR) for evaluation, which are consistent with research works from literature.

**Other Comments Or Suggestions:**

One major suggestion is that the main body of your submission should be self-contained. However, the authors heavily rely on Appendices for details, which makes it inconvenient for readers.

**Other Strengths And Weaknesses:**

Strengths:
S1. The paper is overall well-written with interesting ideas.
S2. Extensive experimental results are presented.
S3. The presentation is clear and precise.

Weaknesses:
W1. The TST module, from my point-of-view, serves as a temporal global feature switch, facilitating interaction between features of different modalities by acting as an intermediate channel. However, the time alignment function of the TST module seems to be duplicated with the positional encoding. In Table 4, the experimental results of different shared token configurations inadvertently undermine the claimed significance of the time-aligned mechanism.
W2. No visualized results or case studies to intuitively demonstrate the model’s effectiveness.
W3. Neither time-complexity analysis nor the computational overhead is presented, raising questions in real-world applicability.
W4. SASRec, BERT4Rec, and MMMLP are all published before 2024. I would like to see comparisons with the latest SOTA methods like DiffMM (https://dl.acm.org/doi/pdf/10.1145/3664647.3681498), PromptMM (https://dl.acm.org/doi/pdf/10.1145/3589334.3645359), and FETTLE (https://dl.acm.org/doi/pdf/10.1145/3626772.3657701).
W5.  In page 15, the authors claim that the enhanced versions of SASRec and BERT4Rec integrate item ID, text, and image features. However, for MTSTRec, two additional sources are incorporated: prompt-text and price. The unfairness in input sources raises questions about experimental settings.

**Questions For Authors:**

I would like to hear from authors how the TST module functions when aligning time-aware features, as in my opinion, it serves more like an intermediate channel. Will the MTSTRec experience a significant performance degradation when removing the positional encoding in Equation (4)?

**Relation To Broader Scientific Literature:**

As discussed in Methods And Evaluation Criteria section, the proposed method is built upon Transformer-like architectures with an TST fusion module. The transformer was first proposed in [1]. Since then, many works ([2], [3], [4]) in RS adopted such an architecture. Also, time-aware recommendations are also well-explored [5], [6].

[1] Vaswani A, Shazeer N, Parmar N, et al. Attention is all you need. NeurIPS 2017.
[2] de Souza Pereira Moreira G, Rabhi S, Lee J M, et al. Transformers4rec: Bridging the gap between nlp and sequential/session-based recommendation. RecSys 2021.
[3] Sun F, Liu J, Wu J, et al. BERT4Rec: Sequential recommendation with bidirectional encoder representations from transformer. CIKM 2019.
[4] Li C, Xia L, Ren X, et al. Graph transformer for recommendation. SIGIR 2023.
[5] Lei Wang, Chen Ma, Xian Wu, et al. Causally Debiased Time-aware Recommendation. WWW 2024.
[6] Qi Zhang, Longbing Cao, et al. Neural time-aware sequential recommendation by jointly modeling preference dynamics and explicit feature couplings. IEEE TNNLS. 2021.

**Theoretical Claims:**

N/A. This is an Application-driven ML paper, and no theoretical claim is presented.

---

> ### Author Rebuttal · Authors · 2025-04-01
>
> Thank you for your thoughtful feedback. We appreciate your recognition of our work’s clarity, the thoroughness of the experimental evaluation, and a well-written presentation with interesting ideas.
>
> Below, we address each comment and concern in detail:
>
>
> [W1] Redundancy between TST and positional encoding; unclear role in time alignment [Weakness] [Questions]
>
> [R1] TST aligns features across modalities at each time step, complementing positional encoding’s intra-modality ordering. Together, they ensure robust temporal alignment and fusion. Omitting positional encoding sometimes yields decent results (e.g., NDCG@5: 0.8946 vs. 0.8942, NDCG@10: 0.9105 vs. 0.9086), due to TST’s implicit time handling. However, retaining it ensures fair Transformer-based comparisons and model stability.
>
>
> [W2] No visualized results or case studies to demonstrate the model’s effectiveness. [Weakness]
>
> [R2] Ｗe provided additional visualized results. See our response to Reviewer RAjD’s comment [R3].
>
>
> [W3] Time-complexity analysis [Weakness]
>
> [R3] MTSTRec’s complexity primarily arises from Transformer layers at $O(n^2 \cdot d)$ per modality, yielding a naive upper bound of $O(m \cdot n^2 \cdot d)$. TST adds just one extra token per time step (plus a prediction token), so the overhead is small. With optimized implementations, MTSTRec remains only a minor constant overhead above standard Transformer-based sequential models. See [R2] in Reviewer RAjD.
>
>
> [W4] Comparisons with the latest SOTA methods like DiffMM, etc. [Weakness]
>
> [R4] Our method emphasizes temporal ordering, whereas DiffMM and PromptMM lack timestamps, complicating comparisons. FETTLE operates more as a plug-in than a standalone framework, though we may use its ideas later. Truly multimodal sequential recommenders remain rare; MMMLP (2023) is a notable example. For fairness, we also adapted SASRec and BERT4Rec to handle multimodal inputs.
>
>
> [W5] Unfair comparison due to different input sources across models [Weakness]
>
> [R5] We used prompt-text and price primarily to showcase TST’s mid-fusion scalability rather than any specialized engineering. For fairness, SASRec+ and BERT4Rec+ use matching text and image inputs. Ablations show MTSTRec (without prompt-text) still surpasses SASRec (e.g., 0.8574 vs. 0.8015 NDCG@5), proving TST’s advantage extends beyond additional modalities. Our core contribution is TST’s effective integration of diverse features, further boosted by extra signals.
>
>
> [W6] Clarification on the contribution of multimodal features vs. ID embeddings [Claims And Evidence]
>
> [R6] We acknowledge that ID embeddings are powerful predictors, yet our ablation studies show that removing other modalities—especially Text and Prompt Text—causes noticeable performance drops, as shown in Table 2 (leading to a significant drop in NDCG@5 from 0.8800 to 0.8574). This highlights the collective benefit of multimodal features.
>
>
> [W7] Lack of non-e-commerce datasets for generalizability [Experimental]
>
> [R7] Suitable non-e-commerce datasets for session-based, multimodal recommendations are hard to find, and time constraints limit broader tests. We are releasing our two private e-commerce datasets and plan to explore additional domains in future work.
>
>
> [W8] Ablation only on one dataset [Experimental]
>
> [R8] To address this concern, we conducted additional ablation studies on the House-Hold E-commerce dataset. The results show consistent performance drops when removing the TST module and fusion encoder, confirming the effectiveness and generalizability of our design across datasets.
> | Fusion Method                                  | NDCG@5 | NDCG@10 | HR@5  | HR@10 | MRR@5 | MRR@10 |
> |------------------------------------------------|--------|---------|-------|--------|--------|---------|
> | **MTSTRec**                                    | 0.8942 | 0.9086  | 0.9067 | 0.9358 | 0.8568 | 0.8607  |
> | w/o TST & Fusion Encoder (Late Fusion)         | 0.8773 | 0.8896  | 0.8839 | 0.9116 | 0.8392 | 0.8429  |
> | w/o Multimodal Encoder (Early Fusion)          | 0.8366 | 0.8516  | 0.8404 | 0.8738 | 0.7929 | 0.7974  |
>
>
> [W9] The method extends established Transformer and time-aware models, and additional related works could be considered for a broader context. [Relation To Literature]
>
> [R9] We have cited [1] and [3], and will add [2], [4], [5], and [6] to better position our method. While based on Transformer, our contribution lies in the TST module, which enables time-aligned multimodal fusion—a key difference from prior works [2–4] that lack such fine-grained alignment. Unlike time-aware methods [5,6] that rely on timestamps, we maintain temporal consistency via position encoding and TST alignment. Our use of prompt-enhanced text and thorough ablations further support the novelty.
>
>
> [W10] Over-reliance on appendix [Comments]
>
> [R10] We agree on the importance of self-containment. However, due to the 8-page limit, we had to move several implementation and analysis details to the Appendix.

---

> > ### Comment · Reviewer_2WPs · 2025-04-03
> >
> > Thanks author for the detailed response, I am inclined to raise my score after carefully reading the rebuttals and comments of other reviewers.

---

> > > ### Author Response · Authors · 2025-04-04
> > >
> > > Thank you so much! We appreciate your positive feedback. All reviewers’ comments helped us improve our paper significantly.

---

### Official Review · Reviewer_RAjD · 2025-03-15

**Overall Recommendation:** 3

**Summary:**

This paper introduces MTSTRec, a multimodal sequential recommendation model that integrates textual, visual, and price information into a unified, time-aligned shared token representation.

**Claims And Evidence:**

The claims made in the paper are generally supported by the evidence.

**Essential References Not Discussed:**

See `Relation To Broader Scientific Literature`.

**Experimental Designs Or Analyses:**

The experiments are well-structured, with ablation studies confirming the importance of each modality.

However, robustness and efficiency evaluations are missing:
- What is the model’s inference speed compared to SASRec?
- How does the model handle noisy or missing modalities?

The comparison is limited.
- LLM-powered recommenders (e.g., LLM-Rec, ACL 2024) should be considered as baselines.

**Methods And Evaluation Criteria:**

The proposed method and evaluation criteria make sense for the problem at hand.

**Other Comments Or Suggestions:**

See `Questions For Authors`.

**Other Strengths And Weaknesses:**

See `Questions For Authors`.

**Questions For Authors:**

1. Introduce random perturbations in images/text to see how robust MTSTRec is to input noise.
2. Report the comparison of training/inference time.
3. Can TST be visualized? Showing attention heatmaps of how tokens interact across modalities could strengthen the argument for TST effectiveness.
4. Would contrastive learning improve TST fusion? Have you considered using multimodal contrastive loss (e.g., CLIP-style loss) to further improve alignment?
5. Missing LLM-based multimodal recommendation baselines.

**Relation To Broader Scientific Literature:**

- The paper builds upon work in multimodal recommendation (e.g., MMMLP, VBPR) and Transformer-based sequential models (e.g., SASRec, BERT4Rec).
- The idea of mid-fusion via shared tokens is related to bottleneck attention mechanisms in multimodal learning (e.g., [1]), and these relevant works should be cited.
- The idea of mid-fusion also relates to Variational Information Bottleneck (e.g., [2, 3]), and these relevant works should be cited.

[1] Nagrani, Arsha, et al. "Attention bottlenecks for multimodal fusion."

[2] Wei, Chunyu, et al. "Contrastive graph structure learning via information bottleneck for recommendation."

[3] Zhao, Wenkuan, et al. "DVIB: Towards Robust Multimodal Recommender Systems via Variational Information Bottleneck Distillation."

**Theoretical Claims:**

The paper does not introduce novel theoretical results but motivates the TST fusion conceptually.

---

> ### Author Rebuttal · Authors · 2025-04-01
>
> Thank you for your thoughtful feedback. We greatly appreciate your recognition of the strengths of our work, particularly your acknowledgment of our conceptual motivation for time-aligned shared token (TST) fusion, the well-structured experiments and ablation studies, and the comprehensive supplementary material.
>
> Below, we address each of your questions in detail.
>
>
> **[Q1]** Robustness and efficiency evaluations [Questions] [Experimental Designs Or Analyses 2]
>
> **[R1]** Thank you for the suggestion. Robustness to noisy inputs is indeed an important aspect of model reliability. While MTSTRec leverages time-aligned shared tokens to fuse modality-specific features—which may inherently help mitigate localized noise—we did not conduct perturbation experiments in this version due to time constraints.
>
>
> **[Q2]** Comparison of training/inference time [Questions] [Experimental Designs Or Analyses 1]
>
> **[R2]** The following table shows that MTSTRec requires more training time (83 minutes) due to its multimodal inputs (ID, text, image, prompt-text, price) and the TST module (Sec. 3.3.2), but its inference time (12.43 seconds) remains competitive—only slightly higher than SASRec+ (7.48 seconds). In return, MTSTRec achieves significantly better accuracy (NDCG@5 = 0.8942 vs. 0.8150 for SASRec+, Sec. 4.2), offering a strong performance–efficiency trade-off suitable for real-world use. We provided the time complexity analysis. See [R3] for Reviewer 2WPs.
> | Model      | Training Time (minutes) | Inference Time (seconds) |
> |----------------|-----------------------------|------------------------------|
> | MTSTRec        | 83                          | 12.43                        |
> | SASRec       | 0.083                          | 1.860                          |
> | BERT4Rec       | 4.1                         | 1.88                        |
> | SASRec+     | 66                          | 7.48                         |
> | BERT4Rec+   | 59                          | 7.89                         |
>
>
>
> **[Q3]** Visualize TST [Questions]
>
> **[R3]** The attention heatmaps from the MTSTRec model at layer 0 visualize the self-attention weights for a sequence of 21 tokens (18 product IDs padded with 2  zeros to reach the maximum length of 20) across five modalities: token (product ID), style (image), text, prompt-text, and sale price. Each 21x21 heatmap shows how much each token attends to others, with the x-axis and y-axis representing sequence positions (0 to 20). Yellow indicates high attention, and dark blue/purple indicates low attention. The heatmaps reveal modality-specific patterns: token focuses on item identity with sparse attention, style captures visual similarities with broader attention, text and prompt emphasize textual features with vertical stripes, and sale price shows scattered attention, leading to less importance in recommendation. https://anonymous.4open.science/r/MTST_ICML_rebuttal-0E5D/MTST_attention_heatmap.png
>
>
> **[Q4]** Potential of contrastive learning (e.g., CLIP-style loss) to improve TST fusion [Questions]
>
> **[R4]** Integrating a multimodal contrastive loss such as CLIP-style alignment is an intriguing direction, particularly for bridging text–image or text–style embeddings. In principle, a contrastive term might further align the modalities during TST fusion, potentially sharpening the shared token’s cross-modal representation. However, we have not explicitly explored a full-blown contrastive learning design so far, primarily because the supervised next-item prediction objective already forces alignment across modalities that correlate to the same product and also because contrastive training can introduce substantial additional computational overhead. That said, exploring a dual-objective setup—where TST’s latent space is refined by a CLIP-like objective—could be a compelling avenue for future research.
>
>
> **[Q5]** Missing LLM-based multimodal recommendation baselines [Questions] [Experimental Designs Or Analyses 3]
>
> **[R5]** We acknowledge recent progress in LLM-based recommenders such as LLM-Rec. While these approaches inspire our work, MTSTRec is designed for a multimodal sequential recommendation, integrating not only text but also images, prices, and item IDs via our proposed TST module. In contrast, LLM-Rec focuses on text-only scenarios with different modeling objectives. It is worth noting that LLMRec can be seen as analogous to our prompt encoder. Therefore, our ablation studies—comparing with early fusion, late fusion, and using only the prompt encoder—can be viewed as an indirect comparison with LLMRec.
>
>
> **[W1]** Missing related work on mid-fusion strategies [Relation To Broader Scientific Literature]
>
> **[R6]** Thank you for the helpful comments. We would like to clarify that [1] has already been cited in our paper. We agree that [2, 3] are relevant to our mid-fusion design and will include proper citations and discussion in the final version.

---

### Official Review · Reviewer_W2Te · 2025-03-16

**Overall Recommendation:** 3

**Summary:**

The authors propose a sequential recommendation framework focusing on multi modal feature fusion. In the proposed models, the authors include feature sets like product IDs, images, text, and prices.The main contribution comes from authors proposing a new block named Time-aligned Shared Token Fusion module. Each modal has its own self-attention block and the TST module learns an element wise average pooling token (z^sh) to be concatenated together with the modal specific token z^mod to be fed into the next layer's input to learn the mod specific patterns for the next layer. This basically forced the model to add a fused cross modal feature input into each layer. The authors conducted offline analysis on 3 datasets, including performance comparison and ablation evaluations on importance of multi-modal features and impact of different ways of multi-modal fusion.

**Claims And Evidence:**

I found the claims of the authors convincing. However, I will be more convinced of the results if there are online performance evaluations of the proposed algorithm (instead of just offline evaluation). I totally understand that not all researchers(especially on recommendation domain) will have access to online experiments, but since the authors are using private data from AviviD Innovative Multimedia for experiment evaluation, it will be great if there is any online results can be shared.

**Essential References Not Discussed:**

No

**Experimental Designs Or Analyses:**

Comprehensive experiments are conducted on 3 offline datasets(1 is public right now and the author promise to make the other 2 public once the paper is published). The experiment design makes sense.

**Methods And Evaluation Criteria:**

The methods makes sense and is easy to follow.

**Other Comments Or Suggestions:**

Please refer to Other Strengths And Weaknesses section. Thanks!

**Other Strengths And Weaknesses:**

Strength

1.Sequential recommendation is a very well established domain and there are many real world applications that can make the work relevant to a large group of audiences.

2. The proposed method makes sense and is actually straightforward and easy to follow. The innovation is not ground-breaking but more incremental on top of existing/well-established techniques. But for recommendation systems, sometimes a not-complicated but working solution is way more important than an over-complicated algorithm design.

3. The experiments are well designed, comprehensive and include all the analysis and ablation I would like to know as a reader. The offline analysis shows significant gains of the proposed algorithm on top of the existing baselines.

4. The paper is well written and easy to follow

5. The authors promised to release 2 of the offline dataset to the public after the paper is published, which can benefit follow-up researches.


Weakness or Questions

1. My biggest concern is the lack of online test results. This is definitely not a reason to reject this paper but this definitely impacts on the confidence of whether this work will really work in real-life settings. In sequential recommendation, there can be a lot of times that offline evaluation results do not match online performance and an online LE can make the conclusion more convincing.
If the authors have access to online evaluation, it will be great if they can share some of the learnings.

2. My second concern is that the proposed model will be much larger than many of the baselines. I.e., the proposed model uses an independent transformer block for each modal and has multiple layers on top of it vs baselines like bert4rec which comes with a much smaller model structure. How much of gain is coming from the model size(e.g. the effective size of the model) and how much is coming from the proposed methods?

Can authors provide the parameter size of different models used in the experiment sections?

Some nit questions(not concerns):

3. Why do the authors add a close embedding token (z_cz) for each modality at the end of each sequence instead of directly leveraging the output of the last activity of the sequence? Does this design choice come with gain observed offline?

4. Why the final output only concatenate all the modal specific tokens (z_mod) btu did not concatenate the shared token (z_sh)?

**Questions For Authors:**

Please refer to Other Strengths And Weaknesses section. Thanks!

**Relation To Broader Scientific Literature:**

N/A

**Theoretical Claims:**

I checked the correctness of the proposed method and they looked correct to the best of my knowledge.

---

> ### Author Rebuttal · Authors · 2025-04-01
>
> We sincerely thank the reviewer for the thoughtful and constructive feedback. We appreciate the recognition of the practicality and clarity of our proposed method, the comprehensiveness of our experiments, and the potential impact of releasing two of our datasets to the public. We are especially grateful for your comment that “a not-complicated but working solution is way more important than an over-complicated algorithm design,” which aligns with our goal of proposing practical and effective methods for real-world recommendation scenarios.
>
> We have carefully addressed the reviewer’s comments and questions as follows.
>
>
> **[W1]** Lack of online testing [Weaknesses]
>
> **[R1]** We completely agree with your comment on online testing. This work is part of an industry-academia collaboration, and the proposed MTSTRec model has so far been developed and evaluated using historical data. As the next step, an online A/B test is planned to assess the model’s performance in a live setting. This paper, if accepted, will serve as strong evidence to convince the top management of the company to go ahead with the online testing. We hope to share what we will learn with the research community soon.
>
>
> **[W2]** Model size vs. performance gain/parameter comparison [Weaknesses]
>
> **[R2]** We acknowledge that MTSTRec has more parameters than the baseline models that are based solely on historical interaction data, such as SASRec and BERT4Rec, due to its modality-specific encoders and fusion layers for handling multimodal inputs. The parameter sizes of the models used in our experiments are:
> * MTSTRec: 59.24M
> * SASRec: 4.69M
> * BERT4Rec: 7.84M
> * SASRec+: 203.67M
> * BERT4Rec+: 400.33M
>
> Although we were unable to obtain the exact parameter sizes for MMMLP and MMMLP+, MTSTRec is significantly smaller than SASRec+ and BERT4Rec+, which serve as enhanced multimodal baselines.
> As shown in our ablation studies, the performance improvements of MTSTRec are not merely due to model size but largely result from the proposed TST fusion module and time-aligned multimodal design.
>
>
> **[Q3]** Why add a $z_{cz}$ token instead of using the last token? [Weaknesses]
>
> **[R3]** We have explored both approaches during development. Our experiments showed that relying solely on the last item token often obscured modality-specific contributions, particularly in short sequences or sessions with abrupt shifts, as it conflated signals across modalities. In contrast, adding a $z_{cz}$ token per modality acts as a dedicated placeholder for “the next item,” aggregating relevant signals from each modality’s perspective and yielding a clearer multimodal representation for prediction. Our testing confirmed that this approach consistently outperformed the last-token method across metrics like NDCG@20, enhancing accuracy without significant computational overhead.
> Due to space constraints, we did not elaborate on this comparison in the original paper. However, we recognize its importance and will include a detailed discussion, along with supporting experimental results, in Section 4 or an appendix of our revised paper to further substantiate this design choice. Thank you for highlighting this point.
>
>
> **[Q4]** Why not concatenate the shared token ($z_{sh}$) at the final output? [ Weaknesses]
>
> **[R4]** We evaluated both approaches—concatenating the shared tokens in the final output and excluding them—and found that preserving only the modality-specific CLOZE tokens yields better performance. The shared tokens effectively align cross-modal features during fusion, but adding them to the final output diluted modality-specific details essential for accurate predictions. We will provide more details on these findings in our revised paper.

---

> > ### Comment · Reviewer_W2Te · 2025-04-01
> >
> > I would like to thanks for the detailed response from the authors. The authors have answered most of my questions.
> >
> > After careful review of the rebuttal and comments from other reviewers, my recommendation score stay the same.

---

> > > ### Author Response · Authors · 2025-04-04
> > >
> > > Thank you for the positive feedback! We would really appreciate it if you could kindly let us know which questions remain unanswered.
> > >
> > > (2025/04/07 update):
> > >
> > > Thank you for your previous review and feedback. Recent updates to our rebuttal have received positive feedback and led to increases in scores from other reviewers. We would be truly grateful if you could kindly revisit our responses. Your further consideration could make a meaningful difference, and we deeply appreciate the time and effort you’ve invested in reviewing our work.

---

### Decision · Program_Chairs · 2025-05-01

**Decision:**

Accept (poster)

**Comment:**

The evaluation from four reviewers reached a borderline. After carefully checking the paper, the reviews, the rebuttal, and the author-reviewer discussions, I think the strong points slightly outweigh the weak points. Thus, it is a weak accept.